# ASYNCHRONOUS DENOISING DIFFUSION MODELS FOR ALIGNING TEXT-TO-IMAGE GENERATION

**Zijing Hu**[1], **Yunze Tong**[1], **Fengda Zhang**[2], **Junkun Yuan**[1], **Jun Xiao**[1], **Kun Kuang**[1*]

[1]Zhejiang University, [2]Nanyang Technological University

`{zj.hu,tyz01}@zju.edu.cn, fdzhang328@gmail.com,`
`yuanjk0921@outlook.com, junx@cs.zju.edu.cn, kunkuang@zju.edu.cn`

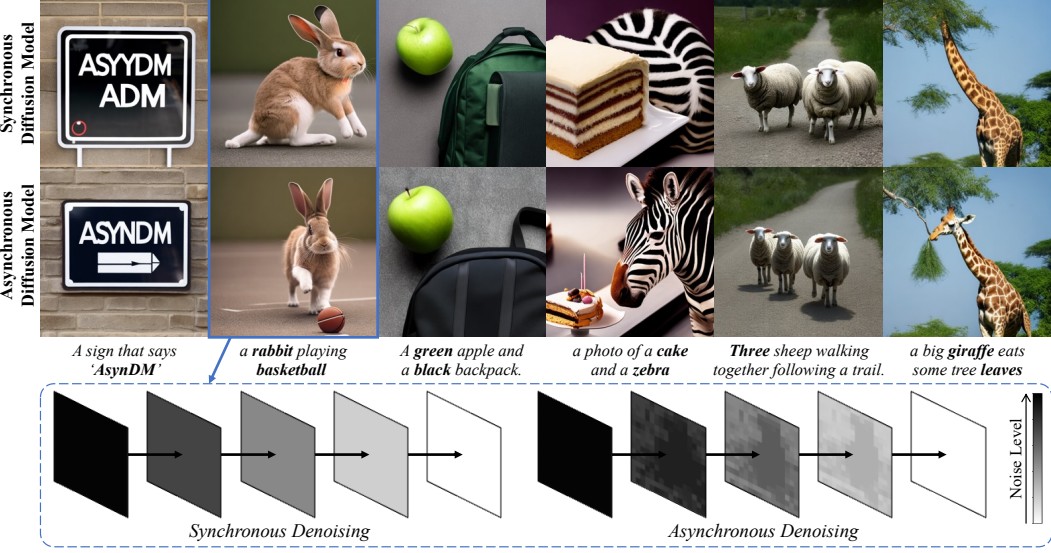

Figure 1: Existing diffusion models generate images through synchronous denoising, where all pixels are simultaneously denoised step-by-step from noises to images, hindering text-to-image alignment. Asynchronous diffusion models denoise the prompt-related regions more gradually than other regions, thereby receiving clearer inter-pixel context and ultimately achieving improved alignment.

## ABSTRACT

Diffusion models have achieved impressive results in generating high-quality images. Yet, they often struggle to faithfully align the generated images with the input prompts. This limitation is associated with synchronous denoising, where all pixels simultaneously evolve from random noise to clear images. As a result, during generation, the prompt-related regions can only reference the unrelated regions at the same noise level, failing to obtain clear context and ultimately impairing text-to-image alignment. To address this issue, we propose asynchronous diffusion models—a novel framework that allocates distinct timesteps to different pixels and reformulates the pixel-wise denoising process. By dynamically modulating the timestep schedules of individual pixels, prompt-related regions are denoised more gradually than unrelated regions, thereby allowing them to leverage clearer inter-pixel context. Consequently, these prompt-related regions achieve better alignment in the final images. Extensive experiments demonstrate that our asynchronous diffusion models can significantly improve text-to-image alignment across diverse prompts. The code repository for this work is available at `https://github.com/hu-zijing/AsynDM`.

---

*Corresponding author.

# 1 INTRODUCTION

Diffusion models have achieved remarkable success across a wide range of domains, such as robotics (Chi et al., 2024; Wolf et al., 2025), classification (Li et al., 2023a; Tong et al., 2025a), image perception and segmentation (Amit et al., 2021; Yuan et al., 2022; 2023), text generation (Austin et al., 2021; Nie et al., 2025), and visual generation (Yang et al., 2023; Wang et al., 2025). Among these, text-to-image generation has emerged as the most widely recognized application, with the generated images demonstrating impressive diversity and high fidelity (Ho et al., 2020; Rombach et al., 2022). Despite their success, even the most advanced diffusion models still struggle with the issue of *text-to-image misalignment* (Hinz et al., 2020; Ramesh et al., 2022; Feng et al., 2023; Chefer et al., 2023), where the generated images often fail to faithfully match the user-provided prompts, for example with respect to text, color, or count, as illustrated in Figure 1.

We argue that misalignment in diffusion models is closely associated with the issue of *synchronous denoising*. That is, under the formulation of a Markov decision process (Ho et al., 2020; Song et al., 2022), all pixels in an image simultaneously evolve from random noise to a clear state, following the same timestep schedule. At each denoising step, pixels interact by leveraging one another as contextual references, ultimately forming a coherent and harmonious image.

Beyond this, an image is composed of diverse regions. Some of these regions correspond directly to the objects described in the prompt, while others serve as background. For aligned generation, prompt-related regions typically demand more gradual refinement to accurately capture fine-grained semantics. In contrast, prompt-unrelated regions involve fewer semantic constraints and mainly provide supporting context, allowing them to be denoised into a clear state relatively quickly. However, synchronous denoising treats all pixels equally, overlooking the heterogeneous nature of different regions. Consequently, these prompt-related regions always rely on other regions at the same noise level for contextual references. This raises the concern that *synchronuous denoising limits the effective utilization of inter-pixel context, and ultimately hinders text-to-image alignment*.

Based on the above motivation, we propose **Asyn**chronous **D**iffusion **M**odels (AsynDM), a plug-and-play and tuning-free framework that reformulates the denoising process of pre-trained diffusion models. Instead of denoising all pixels simultaneously, the asynchronous diffusion model allows different pixels to be denoised according to varying timestep schedules, as shown in Figure 1. In particular, prompt-unrelated regions can be denoised more quickly, while prompt-related regions are denoised more gradually to ensure sufficient refinement for capturing prompt semantics. These clearer unrelated regions prevent noisy and ambiguous context from bringing uncertainty to the related regions (*e.g.*, undetermined style, shape, etc.). As a result, the related regions can better focus on the content specified by the prompt, thereby enhancing text-to-image alignment.

Moreover, we introduce a method that dynamically identifies the prompt-related regions and modulates the timestep schedules along the denoising process. Specifically, the cross-attention modules (Vaswani et al., 2017) in diffusion models encapsulate rich information about the shapes and structures of the generated images. At each denoising step, we can extract a mask from the cross-attention modules, which highlights the objects in the prompt. Guided by this mask, the asynchronous diffusion model adaptively modulates the timestep schedules of different regions. The highlighted regions (*i.e.*, prompt-related regions) are modulated to be denoised more gradually than other regions (*i.e.*, prompt-unrelated regions), thereby receiving clearer inter-pixel context.

We conduct experiments on four sets of commonly used prompts and compare with advanced baselines. The results show that AsynDM can effectively improve text-to-image alignment both qualitatively and quantitatively. Meanwhile, AsynDM maintains comparable sampling efficiency to the vanilla diffusion model, as it only requires the additional encoding of pixel-wise timesteps.

The main contributions of this paper can be summarized as follows: (1) We highlight that synchronous denoising is a primary reason for the text-to-image misalignment in existing diffusion models. (2) We propose asynchronous diffusion models that introduces pixel-level timesteps, and adaptively modulate the timestep schedules of different pixels, to address the above issue. (3) Comprehensive experiments demonstrate that asynchronous diffusion models consistently improve text-to-image alignment across diverse prompts.

# 2 BACKGROUND

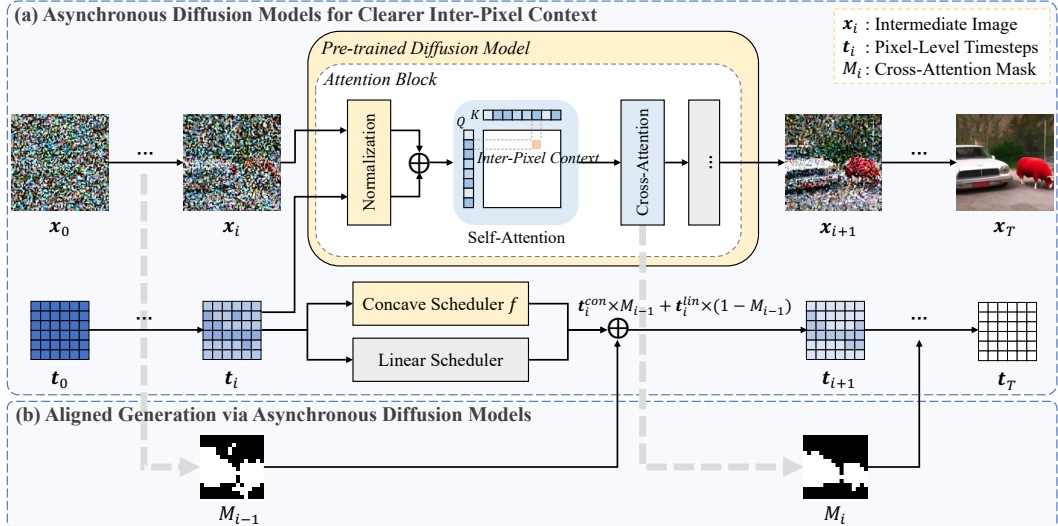

Figure 2: Asynchronous diffusion models improve text-to-image alignment by (a) assigning distinct timesteps to different pixels, where faster-denoised regions provide clearer context, serving as better references for slower ones, and (b) using masks extracted from cross-attention to identify prompt-related regions and dynamically modulate pixel-level timestep schedules.

## 2.1 TEXT-TO-IMAGE DIFFUSION MODELS

**Diffusion Model Formulation.** Diffusion models have emerged as a powerful family of text-to-image generative models. DDPM (Ho et al., 2020) formulates the generation process as a Markovian sequence of latent states. By denoising step by step, these models progressively transform random noise into a coherent image. Based on the DDPM sampler, at each denoising step, the model predicts the last intermediate state $\mathbf{x}_{t-1}$ from current state $\mathbf{x}_t$ according to:

$$p_\theta(\mathbf{x}_{t-1} \mid \mathbf{x}_t, \mathbf{c}) = \mathcal{N}(\mathbf{x}_{t-1} \mid \mu_\theta(\mathbf{x}_t, t, \mathbf{c}), \sigma_t^2 \mathbf{I}), \tag{1}$$

$$\text{with } \mu_\theta(\mathbf{x}_t, t, \mathbf{c}) = \frac{1}{\sqrt{\alpha_t}}(\mathbf{x}_t - \frac{\beta_t}{\sqrt{1 - \bar{\alpha}_t}})\epsilon_\theta(\mathbf{x}_t, t, \mathbf{c}), \tag{2}$$

where $\epsilon_\theta$ denotes the denoising model parimiterized by $\theta$, $\mathbf{c}$ is the prompt, and $\sigma_t$, $\alpha_t$ and $\beta_t$ are timestep-dependent constants. Subsequent extensions, such as DDIM (Song et al., 2022) and DPM-Solver (Lu et al., 2022), further enhance the efficiency and sample quality. These formulations act as the foundation of most modern diffusion-based generative models (Rombach et al., 2022).

**Attention Module in Diffusion Models.** The attention mechanism (Vaswani et al., 2017) has played an important role not only in large language models (Zhao et al., 2025; Han et al., 2025), but also in text-to-image diffusion models (Hertz et al., 2023; Tumanyan et al., 2023). Both UNet-based (Rombach et al., 2022; Podell et al., 2023) and DiT-based (Peebles & Xie, 2023; Esser et al., 2024) diffusion models employ attention blocks to enhance expressiveness. A typical attention block includes a self-attention part and a cross-attention part, and can be formally expressed as:

$$\text{Attention}(Q, K, V) = \text{softmax}(\frac{QK^\top}{\sqrt{d_{\text{key}}}})V, \tag{3}$$

where $Q \in \mathbb{R}^{m \times d_{\text{key}}}$ denotes queries projected from image features, and $K \in \mathbb{R}^{n \times d_{\text{key}}}$, $V \in \mathbb{R}^{n \times d_{\text{value}}}$ denote keys and values, projected either from image features (in self-attention) or from prompt embeddings (in cross-attention). Cross-attention allows the models to condition image generation on textual prompts, while self-attention further enables the models to capture long-range dependencies across the pixels.

## 2.2 DIFFUSION MODEL ALIGNMENT

Text-to-image misalignment has been a longstanding challenge across various generative models, including VAEs, GANs and diffusion models (Zhang & Peng, 2018; Wang et al., 2021; Liao et al.,

2022). Early diffusion model studies have explored methods for conditioning generation on specific factors, such as class labels (Dhariwal & Nichol, 2021; Wang et al., 2023b), styles (Sohn et al., 2023) and layouts (Zheng et al., 2023). The incorporation of text encoders has endowed diffusion models with the capability to generate images from textual descriptions(Rombach et al., 2022). Following this development, recent studies therefore focus on the challenge of text-to-image misalignment in diffusion models, which is essential for the reliable deployment.

On the one hand, researchers have sought to achieve better alignment through fine-tuning. Some studies focus on directly fine-tuning the model (Lee et al., 2023; Tong et al., 2025b), among which reinforcement learning-based methods stand out (Fan et al., 2023; Hu et al., 2025a;b). Others optimize different components without altering the main model parameters. For example, some progressively refine the intermediate noisy images during the denoising process (Chefer et al., 2023; Li et al., 2023c; Rassin et al., 2023), while others optimize prompts to be more precise and informative (Wang et al., 2023a; Mañas et al., 2024).

On the other hand, some studies investigate alignment techniques that do not require fine-tuning. For instance, Z-Sampling (LiChen et al., 2025) enhances alignment by introducing zigzag diffusion step. SEG (Hong, 2024) exploits the energy-based perspective of self-attention to improve image generation. S-CFG (Shen et al., 2024) and CFG++ (Chung et al., 2025) improve text-to-image alignment by refining the classifier-free guidance technique (Ho & Salimans, 2022).

## 3 ASYNCHRONOUS DENOISING FOR CLEARER INTER-PIXEL CONTEXT

In this section, we first introduce the rationale and methodology for allocating distinct timesteps to pixels. We then describe our approach to scheduling the pixel-level timesteps in asynchronous diffusion models. The overview of this section is shown in Figure 2 (a).

### 3.1 PIXEL-LEVEL TIMESTEP ALLOCATION

It is reasonable to allocate distinct timesteps to different pixels. During the denoising process of diffusion models, image features establish inter-pixel dependencies through the attention mechanism, thus pixels can interact with each other and form a coherent image. Notably, timestep information is embedded into the features in a pixel-wise manner external to the attention modules, rather than being directly injected into the attention. In other words, timesteps are involved only in intra-pixel computations, which naturally allows different pixels to be associated with distinct timesteps.

We present the pixel-level timestep formulation of the DDPM sampler, as follows[1]. We adopt $i \in [0, T]$ as the new index of the denoising process, since different pixels have distinct timesteps $t$. This formulation performs denoising from 0 to $T$, rather than from $T$ to 0. Accordingly, the model predicts the next state $\mathbf{x}_{i+1}$ from current state $\mathbf{x}_i$, where $\mathbf{x}_i, \mathbf{x}_{i+1} \in \mathbb{R}^{n_c \times h \times w}$.

$$p_\theta(\mathbf{x}_{i+1} \mid \mathbf{x}_i, \mathbf{c}) = \mathcal{N}(\mathbf{x}_{i+1} \mid \mu_\theta(\mathbf{x}_i, \mathbf{t}_i, \mathbf{c}), \sigma_i^2 \mathbf{I}), \tag{4}$$

$$\text{with } \mu_\theta(\mathbf{x}_i, \mathbf{t}_i, \mathbf{c}) = \frac{1}{\sqrt{\alpha_{\mathbf{t}_i}}}(\mathbf{x}_i - \frac{\beta_{\mathbf{t}_i}}{\sqrt{1 - \bar{\alpha}_{\mathbf{t}_i}}})\epsilon_\theta(\mathbf{x}_i, \mathbf{t}_i, \mathbf{c}), \tag{5}$$

where $\mathbf{t}_i \in \mathbb{R}^{h \times w}$ denotes the timestep states assigned to individual pixels. Specifically, $\alpha_{\mathbf{t}_i}$, $\beta_{\mathbf{t}_i}$ and $\bar{\alpha}_{\mathbf{t}_i}$ denote element-wise indexing, where each entry of $\mathbf{t}_i$ selects corresponding scalar value, yielding matrices of the same shape as $\mathbf{t}_i$. These constant matrices are automatically broadcast along the channel dimension, enabling joint computations with $\mathbf{x}_i$. Moreover, the denoising model $\epsilon_\theta$ can be seamlessly extended to handle pixel-level timesteps by independently encoding them and incorporating the resulting embeddings into the original computation on a per-pixel basis.

The above formulation enables diffusion models to incorporate pixel-level timesteps. Importantly, the asynchronous diffusion model still preserves the *Markov property*. In the asynchronous setting, $\mathbf{t}_i$ becomes a tensor with the same height and width as $\mathbf{x}_i$, serving as a state within the Markov chain, rather than its original role as the reverse-time index.

---

[1]The pixel-level timestep formulation can generalize across diverse diffusion samplers. We also provide the formulation of DDIM sampler in Appendix A.2.

## 3.2 TIMESTEP SCHEDULING IN ASYNCHRONOUS DIFFUSION MODELS

During the denoising process of diffusion models, the noise level of individual pixels gradually decreases as the timestep progresses from $T$ to $0$. In conventional diffusion models, all pixels share the same timestep scheduler from $T$ to $0$, and commonly used samplers, such as DDPM and DDIM, typically implement this progression linearly. In this subsection, we schedule the timesteps and allow certain regions to evolve more slowly than others. This scheduling enables these regions to accumulate clearer inter-pixel context, thereby achieving more gradual refinement.

We adopt the concave function $t = f(i)$ as the scheduler, according to Proposition 1.

**Proposition 1. (See proof in Appendix A.1)** *Let $f(i) : [0, T] \to \mathbb{R}$ be a concave function with $f(0) = T$ and $f(T) = 0$. For any $i_0$ with $0 < i_0 < T$ and any $t_0$ with $T - i_0 \le t_0 \le f(i_0)$, there exist unique constants $a, b$ such that the shifted function $f(i - a) + b$ satisfies:*

$$f(i_0 - a) + b = t_0, \qquad f(T - a) + b = 0. \tag{6}$$

As illustrated in Figure 3, this proposition states that any point located within the shaded area can reach $t = 0$ along the appropriately shifted concave function. In the asynchronous diffusion model, pixels within the target regions (*i.e.*, the prompt-related regions in text-to-image alignment task) are denoised according to the concave function. By applying only a shift to the concave function, regions selected earlier as targets are denoised at a slower rate. Other regions, in contrast, are denoised following a linear function (or a less concave function in some samplers). Therefore, the target regions can be denoised more gradually, thus receive clearer inter-pixel context.

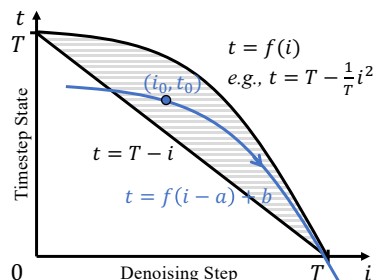

Figure 3: Any point located within the shaded area can reach $t = 0$ along appropriately shifted $f$.

From the perspective of a Markov decision process, in the conventional synchronous diffusion models, the state $\mathbf{x}_t$ transitions to the next state $\mathbf{x}_{t-1}$ under the policy distribution $p_\theta$. Differently, the state in the asynchronous diffusion model is composed of $(\mathbf{x}_i, \mathbf{t}_i)$, which transitions to the next state $(\mathbf{x}_{i+1}, \mathbf{t}_{i+1})$ under the policy distribution $(p_\theta, f)$. In our experiments, we simply adopt a quadratic function $f(i) = T - \frac{1}{T}i^2$ as the scheduling function.

## 4 ALIGNED GENERATION VIA ASYNCHRONOUS DIFFUSION MODELS

In this section, we introduce a method that dynamically identifies the prompt-related regions and modulates the timestep schedules of individual pixels along the denoising process.

**Prompt-Related Region Extraction.** In most text-to-image diffusion models, cross-attention is employed to condition image generation on textual prompts. Even for DiT-based models that rely solely on self-attention, the prompt embeddings are concatenated with image features, thereby enabling implicit cross-attention computations within the self-attention modules (Peebles & Xie, 2023).

In cross-attention computation, the term $\mathrm{softmax}(\frac{QK^\top}{\sqrt{d_{\mathrm{key}}}})$ is commonly referred to as cross-attention maps, denoted by $A \in \mathbb{R}^{|\mathbf{c}| \times h \times w}$, where $|\mathbf{c}|$ is the number of tokens in prompt $\mathbf{c}$. Previous studies (Tang et al., 2023; Hertz et al., 2023; Cao et al., 2023) show that cross-attention maps encapsulate rich information about the shapes and structures of the generated images. Specifically, the $o$-th map in $A$, denoted by $A^o$, highlights the pixels most influenced by the $o$-th token. This property allows us to extract a mask that identifies the image regions most relevant to the prompt, as follows:

$$M = \bigvee_{o \in \mathcal{O}_\mathbf{c}} \{\mathbf{1}[A^o > A^o_{\mathrm{mean}}]\}, \tag{7}$$

where $\mathcal{O}_\mathbf{c}$ denotes the set of token indices corresponding to the objects described in prompt $\mathbf{c}$. For each token $o$, $A^o_{\mathrm{mean}}$ represents the average value of its cross-attention map $A^o$. $\mathbf{1}[\cdot]$ is the indicator function that produces a binary mask based on the given condition, and the operator $\bigvee$ indicates an element-wise logical OR across the resulting masks. This formula ultimately yields a mask that highlights the prompt-related regions.

| DM | DM_concave | Z-Sampling | SEG | SCFG | CFG++ | AsynDM (Ours) |
|---|---|---|---|---|---|---|

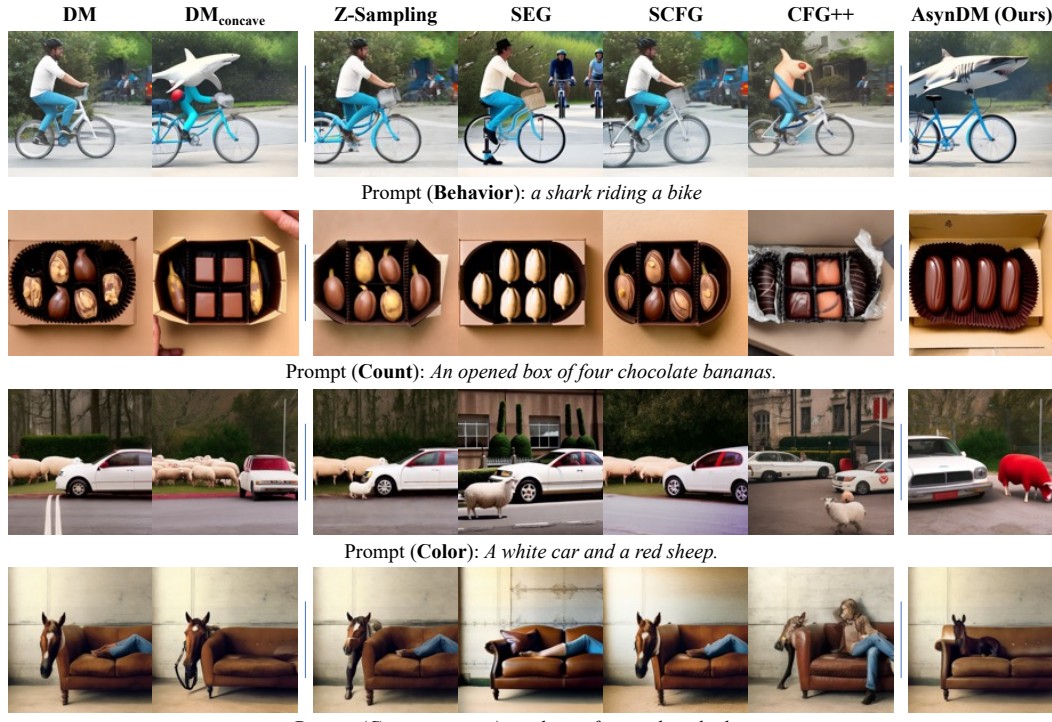

Prompt (**Behavior**): *a shark riding a bike*

Prompt (**Count**): *An opened box of four chocolate bananas.*

Prompt (**Color**): *A white car and a red sheep.*

Prompt (**Co-occurrence**): *a photo of a couch and a horse*

Figure 4: The samples generated by AsynDM and baseline methods across diverse prompts. The images generated by AsynDM show better text-to-image alignment.

**Mask-Guided Asynchronous Denoising.** At each denoising step $i$, we can extract a mask $M_i$ according to Eq.(7). As illustrated in Figure 2 (b), each mask serves as a guidance signal for the next denoising step, where the highlighted regions follow the concave scheduler, and the remaining regions follow the linear scheduler. As denoising progresses, the image gradually becomes clearer in a coarse-to-fine manner (Park et al., 2023; Rissanen et al., 2023), and the mask correspondingly evolves to precisely indicate the shapes and positions of the objects. Consequently, the object-related regions are dynamically modulated to denoise more slowly and gradually, thereby receiving clearer inter-pixel context. The clearer context enables these object-related regions to better focus on the content specified by the prompt, ultimately yielding more faithful and aligned image generation.

## 5 Experiments

In this section, we first introduce our experimental setting. Next, we demonstrate the effectiveness of AsynDM in improving text-to-image alignment, providing both qualitative and quantitative results across diverse prompts and in comparison with multiple baselines. Finally, we conduct ablation on the mask and the concave scheduler, demonstrating the effectiveness and robustness of AsynDM.

### 5.1 Experimental Setting

**Diffusion Models.** We adopt Stable Diffusion (SD) 2.1-512-base (Rombach et al., 2022), one of the commonly used UNet-based diffusion models, as the foundation model of our experiments. The total timesteps $T$ is set to $50$. We employ the DDIM sampler (Song et al., 2022), and the noise weight $\eta$ is set to $1.0$, which determines the extent of randomness at each denoising step. We also conduct experiments on more advanced diffusion models, including the UNet-based SDXL-base-1.0 (Podell et al., 2023) and DiT-based SD3.5-medium (Esser et al., 2024). The experimental results on these models are shown in Appendix D.1.

**Prompts.** We adopt four commonly used prompt sets in our experiments. (1) *Animal activity* (Black et al., 2023). This prompt set has the form "*a(n) [animal] [activity]*", where the activities come from

Table 1: Text-to-image alignment performance of AsynDM compared with baseline methods across diverse prompts.

| Prompt Set | Method | BERTScore↑ | CLIPScore↑ | ImageReward↑ | QwenScore↑ |
|---|---|---|---|---|---|
| Animal Activity | DM | 0.6353 | 0.3685 | 0.7543 | 4.9445 |
| | DM$_{concave}$ | 0.6381 (+0.0028) | 0.3715 (+0.0030) | 0.8544 (+0.1001) | 5.0695 (+0.1250) |
| | Z-Sampling | 0.6353 (+0.0000) | 0.3708 (+0.0023) | 0.8283 (+0.0740) | 5.0242 (+0.0797) |
| | SEG | 0.6309 (-0.0044) | 0.3605 (-0.0080) | 0.6493 (-0.1050) | 4.7632 (-0.1813) |
| | S-CFG | 0.6383 (+0.0030) | 0.3716 (+0.0031) | 0.8653 (+0.1110) | 5.0421 (+0.0976) |
| | CFG++ | 0.6249 (-0.0104) | 0.3565 (-0.0120) | 0.3284 (-0.4259) | 4.4484 (-0.4961) |
| | AsynDM | **0.6414 (+0.0061)** | **0.3750 (+0.0065)** | **0.9219 (+0.1676)** | **5.5218 (+0.5773)** |
| Drawbench | DM | 0.6968 | 0.3659 | 0.3943 | 4.7406 |
| | DM$_{concave}$ | 0.6970 (+0.0002) | 0.3670 (+0.0011) | 0.4152 (+0.0209) | 4.8179 (+0.0773) |
| | Z-Sampling | 0.6979 (+0.0011) | 0.3676 (+0.0017) | 0.4505 (+0.0562) | 4.7656 (+0.0250) |
| | SEG | 0.6925 (-0.0043) | 0.3527 (-0.0132) | 0.2478 (-0.1465) | 4.6695 (-0.0711) |
| | S-CFG | 0.6972 (+0.0004) | 0.3693 (+0.0034) | 0.4398 (+0.0455) | 4.8750 (+0.1344) |
| | CFG++ | 0.6938 (-0.0030) | 0.3539 (-0.0120) | 0.1644 (-0.2299) | 4.6210 (-0.1196) |
| | AsynDM | **0.7007 (+0.0039)** | **0.3701 (+0.0042)** | **0.4560 (+0.0617)** | **4.9804 (+0.2398)** |
| GenEval | DM | 0.7030 | 0.3620 | 0.1541 | 4.9390 |
| | DM$_{concave}$ | 0.7039 (+0.0009) | 0.3637 (+0.0017) | 0.1979 (+0.0438) | 4.9976 (+0.0586) |
| | Z-Sampling | 0.7046 (+0.0016) | 0.3626 (+0.0006) | 0.1757 (+0.0216) | 4.9179 (-0.0211) |
| | SEG | 0.7005 (-0.0025) | 0.3493 (-0.0127) | 0.0689 (-0.0852) | 4.9125 (-0.0265) |
| | S-CFG | 0.7031 (+0.0001) | 0.3630 (+0.0010) | 0.1819 (+0.0278) | 4.8968 (-0.0422) |
| | CFG++ | 0.6992 (-0.0038) | 0.3482 (-0.0138) | -0.1344 (-0.2885) | 4.5835 (-0.3555) |
| | AsynDM | **0.7081 (+0.0051)** | **0.3683 (+0.0063)** | **0.2895 (+0.1354)** | **5.3390 (+0.4000)** |
| MSCOCO | DM | 0.6995 | 0.3388 | 0.2696 | 5.8507 |
| | DM$_{concave}$ | 0.7004 (+0.0009) | 0.3395 (+0.0007) | 0.2917 (+0.0221) | 5.9632 (+0.1125) |
| | Z-Sampling | 0.6999 (+0.0004) | 0.3377 (-0.0011) | 0.2946 (+0.0250) | 5.8289 (-0.0218) |
| | SEG | 0.6952 (-0.0043) | 0.3295 (-0.0093) | 0.1667 (-0.1029) | 5.8320 (-0.0187) |
| | S-CFG | 0.6995 (+0.0000) | 0.3409 (+0.0021) | 0.3316 (+0.0620) | 5.9328 (+0.0821) |
| | CFG++ | 0.6975 (-0.0020) | 0.3348 (-0.0040) | 0.1471 (-0.1225) | 5.6921 (-0.1586) |
| | AsynDM | **0.7055 (+0.0060)** | **0.3420 (+0.0032)** | **0.3339 (+0.0643)** | **6.2601 (+0.4094)** |

humans, such as "*riding a bike*". (2) *Drawbench* (Saharia et al., 2022). This prompt set consists of 11 categories with approximately 200 prompts, including aspects such as color and count. (3) *GenEval* (Ghosh et al., 2023). This prompt set incorporates 553 prompts, including aspects such as co-occurrence, color and count. (4) *MSCOCO* (Lin et al., 2014). This prompt set is derived from the captions of the MSCOCO 2014 validation set and consists of descriptions of real-world images. For each set, we randomly select 40 prompts for our experiments.

**Metrics.** In our experiments, we employ four metrics to evaluate text-to-image alignment. (1) *BERTScore* (Zhang et al., 2020). This metric leverages a multimodal large language model to generate a description for the image, and then employs BERT-based recall to quantify the semantic similarity between the prompt and the generated description. In our implementation, we use Qwen2.5-VL-7B-Instruct (Wang et al., 2024) to generate descriptions and DeBERTa xlarge model (He et al., 2021) to compute similarity. (2) *CLIPScore*. This metric measures the similarity between the text embeddings and image embeddings encoded by CLIP model (Radford et al., 2021). We use ViT-H-14 CLIP model in our implementation. (3) *ImageReward* (Xu et al., 2023). This metric employs a pre-trained model to estimate human preferences, in which alignment serves as a key factor. (4) *QwenScore*. We employ Qwen2.5-VL-7B-Instruct (Wang et al., 2024) to score text-to-image alignment directly, ranging from 0 to 9. The prompts fed to Qwen are provided in Appendix B.4.

**Baselines.** We sample the diffusion model using both the standard scheduler and the concave scheduler, denoted as DM and DM$_{concave}$, respectively. In addition, we compare AsynDM with the most advanced methods, including Z-Sampling (LiChen et al., 2025), SEG (Hong, 2024), S-CFG (Shen et al., 2024) and CFG++ (Chung et al., 2025).

## 5.2 QUALITATIVE EVALUATION

We first provide the qualitative results of AsynDM in comparison with multiple baselines, as shown in Figure 4. We select several representative prompts that encompass object behavior, count, color,

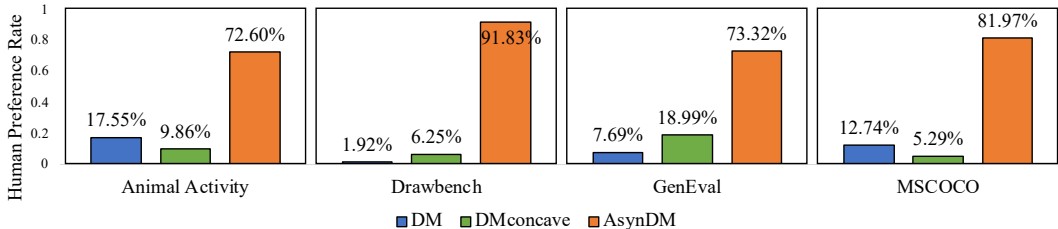

Figure 5: Human preference rates for text-to-image alignment of the images generated by DM, DM$_{concave}$ and AsynDM.

Table 2: Text-to-image alignment performance of AsynDM when employing different concave schedulers and using fixed masks, across prompts from animal activity set.

| Scheduler | Method | BERTScore↑ | CLIPScore↑ | ImageReward↑ | QwenScore↑ |
|---|---|---|---|---|---|
| | DM | 0.6353 | 0.3685 | 0.7543 | 4.9445 |
| Quadratic | DM$_{concave}$ | 0.6381 (+0.0028) | 0.3715 (+0.0030) | 0.8544 (+0.1001) | 5.0695 (+0.1250) |
| | AsynDM | **0.6414 (+0.0061)** | **0.3750 (+0.0065)** | **0.9219 (+0.1676)** | **5.5218 (+0.5773)** |
| | *+fixed mask* | 0.6405 (+0.0052) | 0.3722 (+0.0037) | 0.8642 (+0.1099) | 5.2593 (+0.3148) |
| Piecewise Linear | DM$_{concave}$ | 0.6338 (-0.0015) | 0.3667 (-0.0018) | 0.7043 (-0.0500) | 4.7406 (-0.2039) |
| | AsynDM | **0.6401 (+0.0048)** | **0.3724 (+0.0039)** | **0.8472 (+0.0929)** | **5.2335 (+0.2890)** |
| | *+fixed mask* | 0.6383 (+0.0030) | 0.3705 (+0.0020) | 0.7504 (-0.0039) | 5.0812 (+0.1367) |
| Exponential | DM$_{concave}$ | 0.6352 (-0.0001) | 0.3689 (+0.0004) | 0.7981 (+0.0438) | 4.9289 (-0.0156) |
| | AsynDM | **0.6408 (+0.0055)** | **0.3715 (+0.0030)** | **0.8686 (+0.1143)** | **5.2367 (+0.2922)** |
| | *+fixed mask* | 0.6386 (+0.0033) | 0.3714 (+0.0029) | 0.8374 (+0.0831) | 5.2023 (+0.2578) |

and co-occurrence. The vanilla diffusion model (*i.e.*, DM and DM$_{concave}$) fails to generate images that are well aligned with the prompts. In contrast, AsynDM effectively generates well-aligned images with the same random seeds. Additional qualitative examples, together with those from SDXL and SD 3.5, can be found in Appendix E.

## 5.3 QUANTITATIVE EVALUATION

We also quantitatively demonstrate the text-to-image alignment performance of AsynDM compared with baseline methods. As shown in Table 1, we sample 1,280 images for each of the four prompt sets, using the same random seeds across different methods. The generated images are then evaluated with four metrics. The results demonstrate that AsynDM consistently achieves better alignment across all prompt sets. Meanwhile, sampling 1,280 images takes 78 minutes using the vanilla diffusion model, compared to 86 minutes using AsynDM, which indicates that AsynDM achieves improvements without significantly sacrificing efficiency. In addition, we conduct a human evaluation. We invite 52 participants to choose the image they consider best aligned with the prompt from each group of three candidates, corresponding to DM, DM$_{concave}$ and AsynDM. As shown in Figure 5, the results further demonstrate that AsynDM improves text-to-image alignment.

We also evaluate the image quality of AsynDM using FID-30K (↓). FID-30K refers to the Frechet Inception Distance calculated using 30,000 images from the MSCOCO 2024 validation set as the reference dataset (Pavlov et al., 2023; Lin et al., 2014). We merge all four prompt sets and generate 16,000 images with each of DM, DM$_{concave}$, and AsynDM. The resulting FID-30K scores are 48.63 for DM, 49.29 for DM$_{concave}$, and 49.38 for AsynDM. These results indicate that our method can largely preserve the image quality of the pretrained diffusion model.

## 5.4 ABLATION STUDY

**Ablation on Mask.** In this ablation study, we replace the dynamically updated mask with a fixed mask. This fixed mask is extracted from the average cross-attention map of DM during its denoising process, following Eq.(7). Due to the use of the same random seed, the mask derived from DM can *roughly* highlight the prompt-related regions in the image generated by AsynDM. The results are

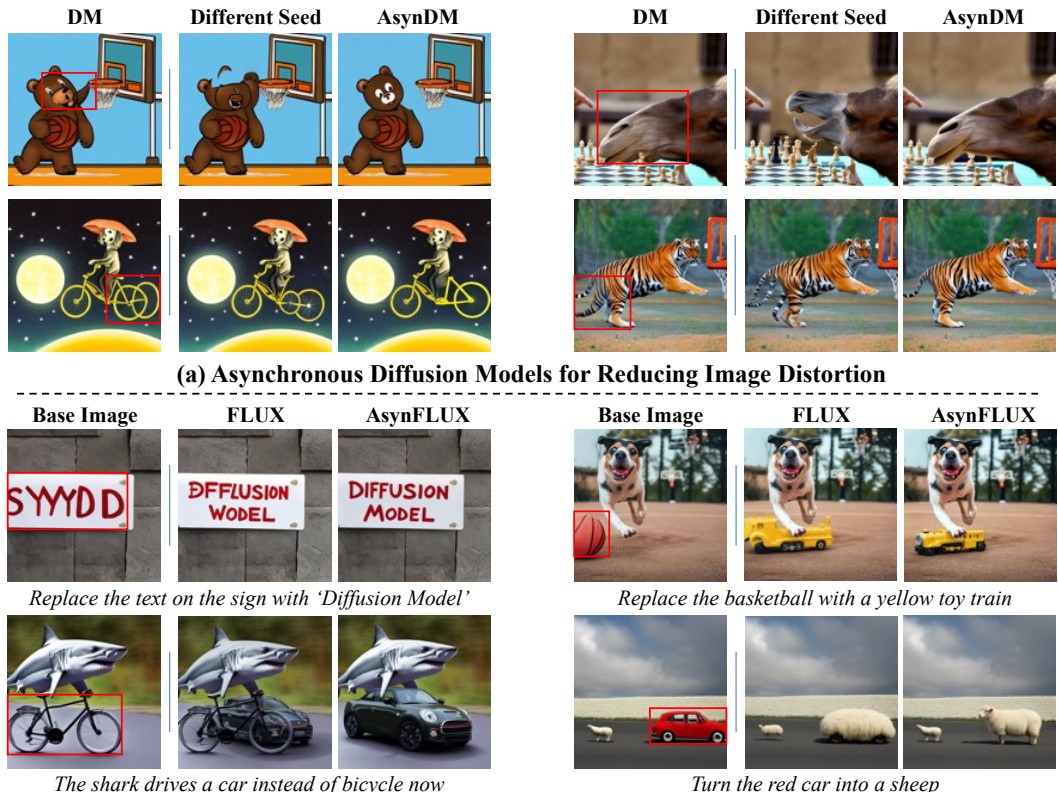

Figure 6: We further employ AsynDM to reduce image distortion and enhance editing performance.

shown in the Table 2. Despite the fixed mask being imperfect, AsynDM still improves text-to-image alignment compared with the base model, demonstrating its robustness to inaccurate masks.

**Ablation on Concave Scheduler.** In addition to the quadratic scheduler, we also employ the piecewise linear scheduler and the exponential scheduler to AsynDM, as follows:

$$f(i) = \min(T - \frac{1}{2}i, \frac{3}{2}T - \frac{3}{2}i), \quad \text{(Piecewise Linear Scheduler)}$$

$$f(i) = \frac{T}{e-1}(e - e^{\frac{1}{T}i}). \quad \text{(Exponential Scheduler)}$$

As shown in Table 2, AsynDM consistently improves image alignment across different schedulers. This is because, across all the variants, these concave schedulers enable the prompt-related regions to receive clearer inter-pixel context. These results further demonstrate the effectiveness and robustness of AsynDM. The image samples of these two ablation studies are provided in Appendix E.

# 6 FURTHER EXPLORATION AND DISCUSSION

**Asynchronous Diffusion Models for Reducing Image Distortion.** Diffusion-generated images often suffer from distortions, such as abnormal limb shapes. As shown in Figure 6 (a), inpainting the distorted regions under different random seeds yields limited improvements. In contrast, applying AsynDM with a mask over the distorted regions, while using the same seed, generates improved images. This suggests that AsynDM has the potential to mitigate image distortions.

**Asynchronous Diffusion Models for Enhancing Editing Performance.** FLUX.1 Kontext is a DiT-based diffusion model that unifies image generation and editing (Labs et al., 2025). However, as shown in Figure 6 (b), even this advanced model can produce edits that mismatch the user prompts. By manually annotating the regions to be edited and applying the concave scheduler during the

editing process, the resulting images align more closely with user expectations. This observation suggests that AsynDM has the potential to further enhance the performance of image editing models.

**Limitations and Future Work.** (1) In this work, we employ a fixed concave function to guide the transition of timestep states. A promising direction for future research is to replace this fixed function with a learnable model that can adaptively predict the next timestep state for each pixel (*e.g.*, Ye et al. (2025); Li et al. (2023b)), potentially leading to more flexible and accurate transitions. (2) We only distinguish between prompt-related and unrelated regions. A natural extension would be to capture more complex object relationships by sorting the objects or constructing a directed acyclic graph (Han et al., 2024; Kong et al., 2025). Assigning different objects with varying concave schedulers may further lead to improved performance. (3) When timestep states across pixels differ extremely, the faster denoised regions may be affected by noisy regions, causing the final image to retain a considerable amount of noise (See Appendix D.2 for an example). We attribute this limitation to the training-free nature of AsynDM, which makes it less robust to large disparities in noise levels. Future work could address this issue through fine-tuning or pre-training.

## 7 CONCLUSION

In this work, we propose the asynchronous denoising diffusion models to improve text-to-image alignment. The AsynDM allocates distinct timesteps to individual pixels and schedules them using a concave function. Guided by the masks that highlight the prompt-related regions, these regions can be denoised more slowly than unrelated ones, allowing them to receive clearer inter-pixel context. The clearer context can help the related regions better capture the content specified by the prompts, thereby generating more aligned images. Our empirical results demonstrate the effectiveness and robustness of the proposed asynchronous diffusion models.

## ACKNOWLEDGEMENTS

This work was supported in part by the National Natural Science Foundation of China (62376243), the National Key Research and Development Program of China (2024YFE0203700), and "Pioneer" and "Leading Goose" R&D Program of Zhejiang (2025C02037). All opinions in this paper are those of the authors and donot necessarily reflect the views of the funding agencies.

## ETHICS STATEMENT

This paper aims to advance the broader field of text-to-image alignment in diffusion models. While our method focuses on improving controllability and semantic faithfulness in generative models, it may have societal implications similar to those associated with image synthesis technologies. We do not identify any concerns unique to our approach that require special emphasis. For a more extensive discussion of the ethical considerations and broader impacts surrounding diffusion models and text-to-image generation, we refer interested readers to Po et al. (2024).

## REPRODUCIBILITY STATEMENT

We have taken several measures to ensure the reproducibility of our work. The detailed experimental setting is provided in Section 5.1 of the main paper, and the Appendix B includes comprehensive implementation details, such as hyperparameters. To further ensure reproducibility, we provide pseudo-code that outlines the proposed method step by step in Appendix C.

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

The **Appendix** is organized as follows:

- **Appendix A:** provides the proof of the proposition in the main text and the formulation of the DDIM sampler.
- **Appendix B:** provides more details on implementation.
- **Appendix C:** provides the pseudo-code of employing AsynDM to generate images.
- **Appendix D:** presents more experimental results.
- **Appendix E:** presents more image samples generated by AsynDM.
- **Appendix F:** describes the role of the large language models (LLMs) in preparing this paper.

## A  THEORETICAL DERIVATIONS

### A.1  PROOF OF PROPOSITION 1

From the second equation in Eq.(6) we obtain $b = -f(T - a)$. Substituting into the first equation yields the single-variable condition $f(i_0 - a) - f(T - a) = t_0$. Define:

$$g(a) = f(i_0 - a) - f(T - a), \qquad a \in [0, i_0]. \tag{8}$$

The domain $[0, i_0]$ ensures that both $i_0 - a$ and $T - a$ lie in $[0, T]$.

Since $f$ is concave on $[0, T]$, then $f$ is continuous, hence $g$ is continuous on $[0, i_0]$. Moreover, concavity implies that the slope of $f$ is nonincreasing, which in turn gives:

$$g'(a) = f'(T - a) - f'(i_0 - a) \leq 0, \tag{9}$$

whenever $f$ is differentiable. Therefore, $g$ is nonincreasing on $[0, i_0]$, and strictly decreasing unless $f$ is linear.

At the endpoints, we have:

$$g(0) = f(i_0) - f(T) = f(i_0), \qquad g(i_0) = f(0) - f(T - i_0) = T - f(T - i_0). \tag{10}$$

Therefore, the range of $g$ is exactly the interval $[T - f(T - i_0), f(i_0)]$.

Moreover, since $f$ is concave on $[0, T]$, then:

$$f(T - i_0) = f(\frac{i_0}{T} \cdot 0 + \frac{T - i_0}{T} \cdot T) \geq \frac{i_0}{T} \cdot f(0) + \frac{T - i_0}{T} \cdot f(T) = i_0. \tag{11}$$

Hence $T - f(T - i_0) \leq T - i_0$.

According to the *intermediate value theorem*, for any $t_0 \in [T - i_0, f(i_0)]$, there exists some $a \in [0, i_0]$, such that $g(a) = t_0$. Monotonicity of $g$ guarantees that this solution is unique. Finally, since $a$ is uniquely determined, then $b = -f(T - a)$ is also uniquely determined.

Therefore, the constants $a, b$ exist and are unique.

### A.2  ASYNCHRONOUS DENOISING WITH DDIM SAMPLER

The vanilla DDIM sampler predicts next intermediate state $\mathbf{x}_{t-1}$ according to:

$$\mathbf{x}_{t-1} = \sqrt{\alpha_{t-1}} \cdot \hat{\mathbf{x}}_0 + \sqrt{1 - \alpha_{t-1} - \sigma_t^2} \cdot \epsilon_\theta(\mathbf{x}_t, t, \mathbf{c}) + \sigma_t \epsilon_t, \tag{12}$$

$$\text{with } \hat{\mathbf{x}}_0 = \frac{1}{\sqrt{\alpha_t}}(\mathbf{x}_t - \sqrt{1 - \alpha_t} \cdot \epsilon_\theta(\mathbf{x}_t, t, \mathbf{c})), \tag{13}$$

where $\epsilon_t \sim \mathcal{N}(\mathbf{0}, \mathbf{I})$. The pixel-level timestep formulation of the DDIM sampler is given as follow:

$$\mathbf{x}_{i+1} = \sqrt{\alpha_{\mathbf{t}_{i+1}}} \cdot \hat{\mathbf{x}}_0 + \sqrt{1 - \alpha_{\mathbf{t}_{i+1}} - \sigma_i^2} \cdot \epsilon_\theta(\mathbf{x}_i, \mathbf{t}_i, \mathbf{c}) + \sigma_i \epsilon_i, \tag{14}$$

$$\text{with } \hat{\mathbf{x}}_0 = \frac{1}{\sqrt{\alpha_{\mathbf{t}_i}}}(\mathbf{x}_i - \sqrt{1 - \alpha_{\mathbf{t}_i}} \cdot \epsilon_\theta(\mathbf{x}_i, \mathbf{t}_i, \mathbf{c})), \tag{15}$$

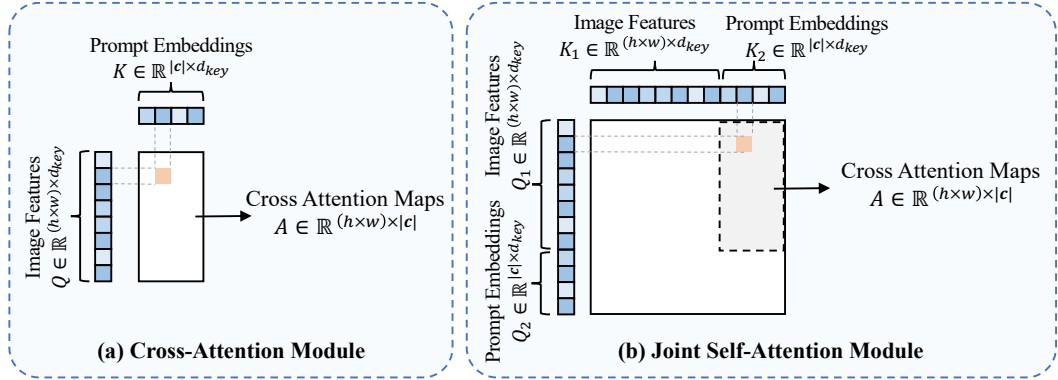

Figure 7: Extracting cross-attention masks from attention modules.

## B IMPLEMENTATION DETAILS

### B.1 IMPLEMENTATION DETAILS OF ASYNDM

**Mask Extraction.** In Section 4, we have described how to extract prompt-related regions from cross-attention maps. However, a model typically contains multiple cross-attention layers, each producing its own set of attention maps. For DiT-based diffusion models, we average the cross-attention maps across all layers and then extract the mask following the procedure outlined in Section 4. In contrast, UNet-based diffusion models comprise layers with varying spatial resolutions. Let $h \times w$ represent the image resolution of $\mathbf{x}_t$, and $h_l \times w_l$ represent the resolution at layer $l$ of the UNet. Inspired by prior work (Hertz et al., 2023; Cao et al., 2023), we only use the cross-attention maps from layers at resolution $h_l \times w_l = \frac{h}{4} \times \frac{w}{4}$. The maps from these layers are averaged to obtain the mask, and subsequently upsampled to the resolution $h \times w$.

**Scheduler Reweighting.** As shown in Appendix D.2, when timestep states across pixels differ extremely, the prompt-unrelated regions in the final image might retain a considerable amount of noise. Therefore, constraining the maximum disparity of timestep states across pixels is fundamental to ensuring that any concave function can be reliably applied for denoising. To achieve this, we adopt a straightforward yet effective strategy by weighting the concave function $f$ with the standard denoising function $g$ (*e.g.*, the linear function). Consequently, the concave function employed for state transitions becomes $f' = \omega \cdot f + (1 - \omega) \cdot g$, where $\omega \in (0, 1)$. The function $f'$ not only retains the concavity, but also mitigates its maximum disparity with respect to the standard function.

### B.2 DETAILS OF HUMAN EVALUATION

The human study was conducted with 52 participants from eight universities, including 23 females and 29 males, whose academic backgrounds ranged from undergraduates to Ph.D. students. Each participant was asked to complete a form. At the beginning of the form, we provided the following instruction: *"Image–prompt alignment refers to how well an image matches the given textual description. For each group of three images, please select the one you believe best matches the given textual description."* The form contained 64 groups of images in total, corresponding to four prompt sets, each comprising 16 groups. These 64 groups were randomly selected from the images sampled during the evaluation of results reported in Table 1. Each group consisted of three images generated by DM, $DM_{concave}$ and AsynDM under the same random seed, accompanied by the corresponding text prompt used for generation. All participants received the same set of images, but the presentation order was randomized, ensuring that participants were unaware of which method each image originated from. The entire form was presented in English.

### B.3 EXTRACTING CROSS-ATTENTION MASKS FROM DIT-BASED MODELS

As shown in Figure 7 (a), in the cross-attention modules, we first obtain the cross-attention maps directly via $A = \text{softmax}(\frac{QK^\top}{\sqrt{d_{\text{key}}}})$, and subsequently derive the corresponding masks using Eq.(7).

However, DiT-based diffusion models typically do not include dedicated cross-attention modules. Instead, they rely on implicit cross-attention computation within the self-attention modules to enable the image to be guided by the prompt. As illustrated in Figure 7 (b), the queries $Q$ and keys $K$ are formed by concatenating the image features with the prompt embeddings. During the attention operation, the resulting attention maps $A_{\text{joint}}$ has a size of $(h \times w + |\mathbf{c}|) \times (h \times w + |\mathbf{c}|)$. By extracting the submatrix corresponding to the interactions between the image-feature queries and the prompt-embedding keys, we can obtain the cross-attention maps (*i.e.*, $A = A_{\text{joint}}[: (h \times w), (h \times w) :]$). The cross-attention masks are then computed using Eq.(7) [2].

## B.4 PROMPT FOR QWEN

We employ Qwen2.5-VL-7B-Instruct (Wang et al., 2024) to score text-to-image alignment with the following prompt: "*You are given an image and a description. Please evaluate how well the image matches the description on a scale from 0 to 9, where 0 means completely unrelated and 9 means perfectly aligned. Return only the score as a single integer without explanation.\n Description: [prompt used to generate the image]*".

## B.5 EXPERIMENTAL RESOURCES

The experiments were conducted on 24GB NVIDIA 3090 GPUs. It tooks approximately 78 minutes for the vanilla diffusion model (SD2.1-512-base) to generate 1,280 images, and approximately 86 minutes for the asynchronous diffusion model.

## B.6 HYPERPARAMETERS

The full hyperparameter list of our experiments is presented in Table 3.

Table 3: Hyperparameters of our experiments.

|  | **Patameter** | **Value** |
|---|---|---|
| Sampling | Denoising steps $T$ | 50 |
|  | Noise weight $\eta$ | 1.0 |
|  | Classifier-free guidance | True |
|  | Guidance scale | 5.0 |
|  | Batch size | 8 |
|  | Batch count | 160 |
| Z-Sampling | Inversion guidance $\gamma_2$ | 0.0 |
|  | Zigzag steps | 49 |
|  | Number of rounds $T_{max}$ | 1 |
| SEG | SEG guidance $\gamma_{seg}$ | 3.0 |
|  | Blurred weight $\sigma$ | 1.0 |
| CFG++ | CFG++ guidance $\lambda$ | 0.4 |

## C PSEUDO-CODE

The pseudo-code of employing the asynchronous diffusion model to generate text-aligned images is shown in Algorithm 1.

---

[2]The cross-attention maps $A$ has a size of $(h \times w) \times |\mathbf{c}|$. The dimension $|\mathbf{c}| \times h \times w$ mentioned in the main text corresponds to its transposed and reshaped form, which is presented to facilitate clearer understanding for the readers.

---

**Algorithm 1:** Pseudo-code of employing the asynchronous diffusion model to generate text-aligned images.

---

**Input** : Total denoising timesteps $T$, number of samples $N$, prompt list $C$, pre-trained diffusion model $\epsilon_\theta$, linear/standard scheduler $g$, concave scheduler $f$.

$D_{sample} = [\,]$ ;
**for** $n \leftarrow 0$ **to** $N - 1$ **do**

    $\mathbf{c} \leftarrow C_n$ ;
    // Initialize $\mathbf{x}_i$, $\mathbf{t}_i$ and $M$
    Randomly choose $\mathbf{x}_0$ from $\mathcal{N}(\mathbf{0}, \mathbf{I})$ ;
    $\mathbf{t}_0 \leftarrow \text{tensor}(\text{shape}(\mathbf{x}_0), \text{fill} = T)$ ;
    $M \leftarrow \text{tensor}(\text{shape}(\mathbf{x}_0), \text{fill} = 1)$ ;
    **for** $i \leftarrow 0$ **to** $T - 1$ **do**

        // Transition of $\mathbf{t}_i$
        $\mathbf{t}_{i+1}^{lin} \leftarrow$ Calculate the next state of $\mathbf{t}_i$ using $g$ ;
        $\mathbf{t}_{i+1}^{con} \leftarrow$ Calculate the next state of $\mathbf{t}_i$ using $f$ ;
        $\mathbf{t}_{i+1} \leftarrow M \times \mathbf{t}_{i+1}^{con} + (1 - M) \times \mathbf{t}_{i+1}^{lin}$;
        // Transition of $\mathbf{x}_i$
        $\epsilon \leftarrow \epsilon_\theta(\mathbf{x}_i, \mathbf{t}_i, \mathbf{c})$, and extract the cross-attention map $A$ ;
        Calculate $\mathbf{x}_{i+1}$ according to the chosen sampler (*e.g.*, Eq.(4) for DDPM) ;
        // Update $M$
        Update $M$ using Eq.(7) ;
    **end**
    $D_{sample}$.append($\mathbf{x}_T$) ;
**end**
**Output:** $D_{sample}$

---

# D  MORE EXPERIMENTAL RESULTS

## D.1  EXPERIMENTS ON SDXL AND SD3.5

We also quantitatively demonstrate the text-to-image alignment performance of AsynDM compared with baseline methods on SDXL and SD 3.5, as shown in Table 4 and Table 5 respectively. For experiments conducted on SD 3.5, we have not included comparisons with Z-Sampling or CFG++. This is because Z-Sampling relies on DDIM inversion, and CFG++ makes modifications to DDIM. However, SD 3.5 is a flow model that is not directly compatible with the DDIM sampler. The experimental results demonstrate that AsynDM consistently achieves better alignment across all prompt sets. The image samples for these experiments are shown in Figure 12 and Figure 13.

Table 4: Text-to-image alignment performance of AsynDM compared with baseline methods on animal activity prompt set. The base model is SDXL-base-1.0 (Podell et al., 2023).

| Prompt Set | Method | BERTScore↑ | CLIPScore↑ | ImageReward↑ | QwenScore↑ |
|---|---|---|---|---|---|
| | DM | 0.6671 | 0.3976 | 1.6552 | 6.6562 |
| | DM$_{concave}$ | 0.6695 (+0.0024) | 0.3993 (+0.0017) | 1.6768 (+0.0216) | 6.8421 (+0.1859) |
| | Z-Sampling | 0.6674 (+0.0003) | 0.4022 (+0.0046) | 1.6677 (+0.0125) | 6.7320 (+0.0758) |
| Animal Activity | SEG | 0.6673 (+0.0002) | 0.3963 (-0.0013) | 1.6417 (-0.0135) | 6.8085 (+0.1523) |
| | S-CFG | 0.6670 (-0.0001) | 0.3981 (+0.0005) | 1.6481 (-0.0071) | 6.6367 (-0.0195) |
| | CFG++ | 0.6581 (-0.0090) | 0.3879 (-0.0097) | 1.3748 (-0.2804) | 6.4078 (-0.2484) |
| | **AsynDM** | **0.6829 (+0.0158)** | **0.4026 (+0.0050)** | **1.6893 (+0.0341)** | **7.2781 (+0.6219)** |

## D.2  ABLATION ON MAXIMUM TIMESTEP DIFFERENCE

Given an extreme concave scheduler $f(i) = \min(T, 2T - 2i)$ and a standard linear scheduler $g(i) = T - i$, the maximum timestep difference between pixels within the same denoising step can reach $\frac{T}{2}$. By interpolating the two schedulers as $f' = \omega \cdot f + (1 - \omega) \cdot g$, we obtain a concave scheduler whose

Table 5: Text-to-image alignment performance of AsynDM compared with baseline methods on animal activity prompt set. The base model is SD3.5-medium (Esser et al., 2024).

| Prompt Set | Method | BERTScore↑ | CLIPScore↑ | ImageReward↑ | QwenScore↑ |
|---|---|---|---|---|---|
| Animal Activity | DM | 0.6590 | 0.3906 | 1.5091 | 6.8812 |
| | DM$_{concave}$ | 0.6603 (+0.0013) | 0.3928 (+0.0022) | 1.6385 (+0.1294) | 7.0656 (+0.1844) |
| | SEG | 0.6570 (-0.0020) | 0.3740 (-0.0166) | 1.4022 (-0.1069) | 7.1250 (+0.2438) |
| | S-CFG | 0.6629 (+0.0039) | 0.3908 (+0.0002) | 1.6227 (+0.1136) | 7.0125 (+0.1313) |
| | AsynDM | **0.6663 (+0.0073)** | **0.3941 (+0.0035)** | **1.6418 (+0.1327)** | **7.2171 (+0.3359)** |

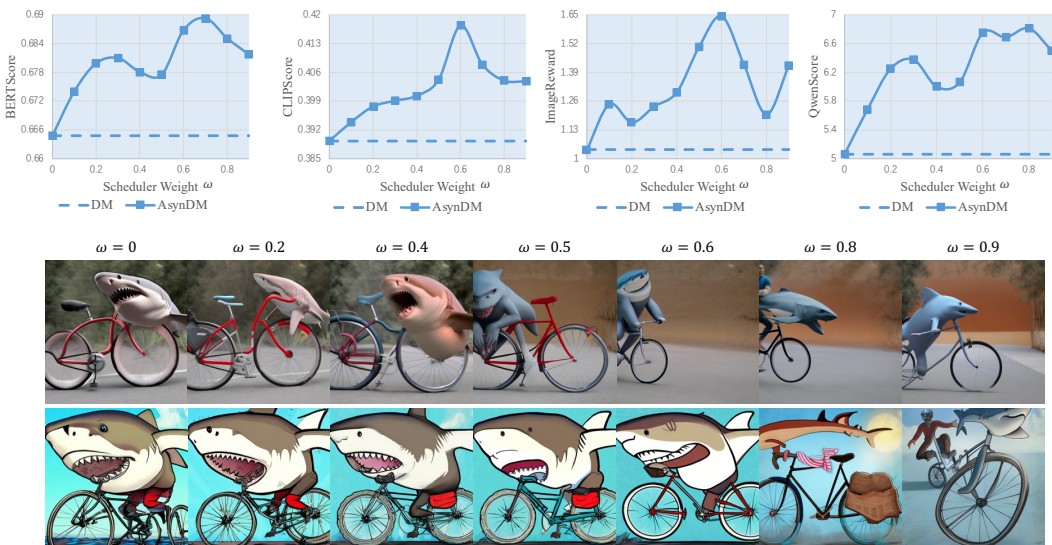

Figure 8: As $\omega$ increases, the maximun timestep difference increases, and the alignment first improves and then degrades. The extreme differences cause faster-denoised regions to retain noise for contextual consistency, leading to blurry and noisy background in final images (*e.g.*, $\omega = 0.8, 0.9$).

maximum timestep difference can be flexibly controlled. As a case study, we consider the prompt "*a shark riding a bike*", and sample 32 images for each value of $\omega$ to evaluate text-to-image alignment. As shown in Figure 8, the results indicate that as $\omega$ increases (*i.e.*, the maximum timestep difference increases), the alignment first improves and then degrades. The degradation occurs because, when timestep states across pixels differ extremely, the faster denoised regions may be affected by noisy regions, which continue to provide noisy context even at later denoising steps. Consequently, these faster denoised regions tend to preserve a considerable amount of noise in order to remain consistent with the context. This effect is particularly evident at $\omega = 0.8$ and $\omega = 0.9$, where the generated images exhibit blurry and noisy background regions.

## D.3 ABLATION ON DENOISING STEPS

A growing body of work has focused on enabling diffusion models to generate high-quality images with only a small number of denoising steps (Xiao et al., 2022; Yin et al., 2024). Motivated by this line of research, we further evaluate the performance of AsynDM under different total denoising steps $T$. Specifically, we set the steps $T$ to 5, 10, 20, 30, 40, 50 and 60, and generate 1,280 images for each setting on the animal activity prompt set. The results are summarized in Table 6. Across all denoising-step configurations, AsynDM consistently improves text-to-image alignment. Figure 9 provides some examples. These results further demonstrate the effectiveness of our method.

Table 6: Alignment performance of AsynDM for different denoising steps $T$, across prompts on animal activity prompt set. The base model is SD2.1-512-base.

| Metric | Method | $T = 5$ | $T = 10$ | $T = 20$ | $T = 30$ | $T = 40$ | $T = 50$ | $T = 60$ |
|---|---|---|---|---|---|---|---|---|
| BERTScore↑ | DM | 0.5924 | 0.6221 | 0.6311 | 0.6330 | 0.6371 | 0.6353 | 0.6346 |
| | AsynDM | **0.5987** | **0.6280** | **0.6364** | **0.6372** | **0.6402** | **0.6414** | **0.6412** |
| CLIPScore↑ | DM | 0.3111 | 0.3574 | 0.3681 | 0.3672 | 0.3689 | 0.3685 | 0.3691 |
| | AsynDM | **0.3260** | **0.3636** | **0.3708** | **0.3699** | **0.3729** | **0.3750** | **0.3752** |
| ImageReward↑ | DM | -1.0882 | 0.0926 | 0.5801 | 0.6458 | 0.7606 | 0.7543 | 0.7668 |
| | AsynDM | **-0.7556** | **0.3087** | **0.6732** | **0.7561** | **0.8692** | **0.9219** | **0.9123** |
| QwenScore↑ | DM | 2.8703 | 4.0750 | 4.7148 | 4.7359 | 4.8992 | 4.9445 | 4.9382 |
| | AsynDM | **3.3164** | **4.5718** | **4.9976** | **5.0218** | **5.2859** | **5.5218** | **5.3234** |

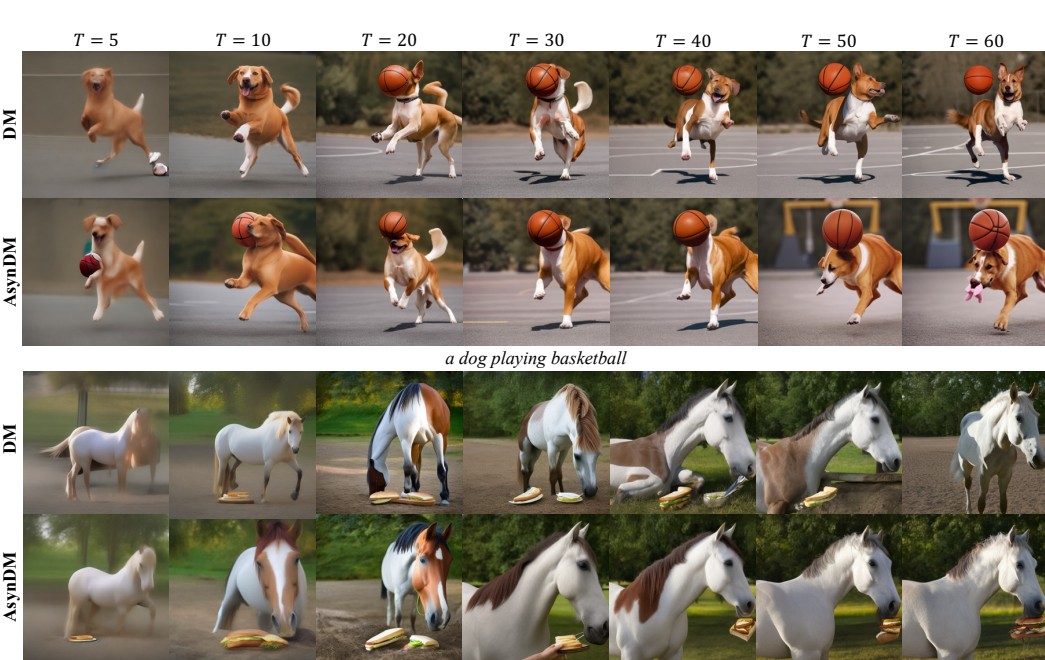

Figure 9: Samples generated by AsynDM compared with DM for different denoising steps $T$. The base model is SD2.1-512-base.

## E  MORE SAMPLES

In this section, we present additional samples generated by AsynDM, alongside those from baseline methods. Specifically, Figure 10 presents more samples on SD 2.1 across diverse prompts. Figure 11 presents the samples of the ablation studies in Section 5.4. Figure 12 and Figure 13 present the samples on SDXL and SD 3.5, respectively.

## F  DECLARATION OF LLM USAGE

In preparing this manuscript, we used the large language model (LLM) as a general-purpose writing assistant. Specifically, the LLM was employed to (1) check grammar and correctness of the text, and (2) suggest more natural and fluent wording. When using the LLM, we first wrote an initial draft of the sentence, and then asked the LLM to check and polish it. The LLM did not contribute to research ideas, methods, experiments, or results. The authors take full responsibility for the content of this paper.

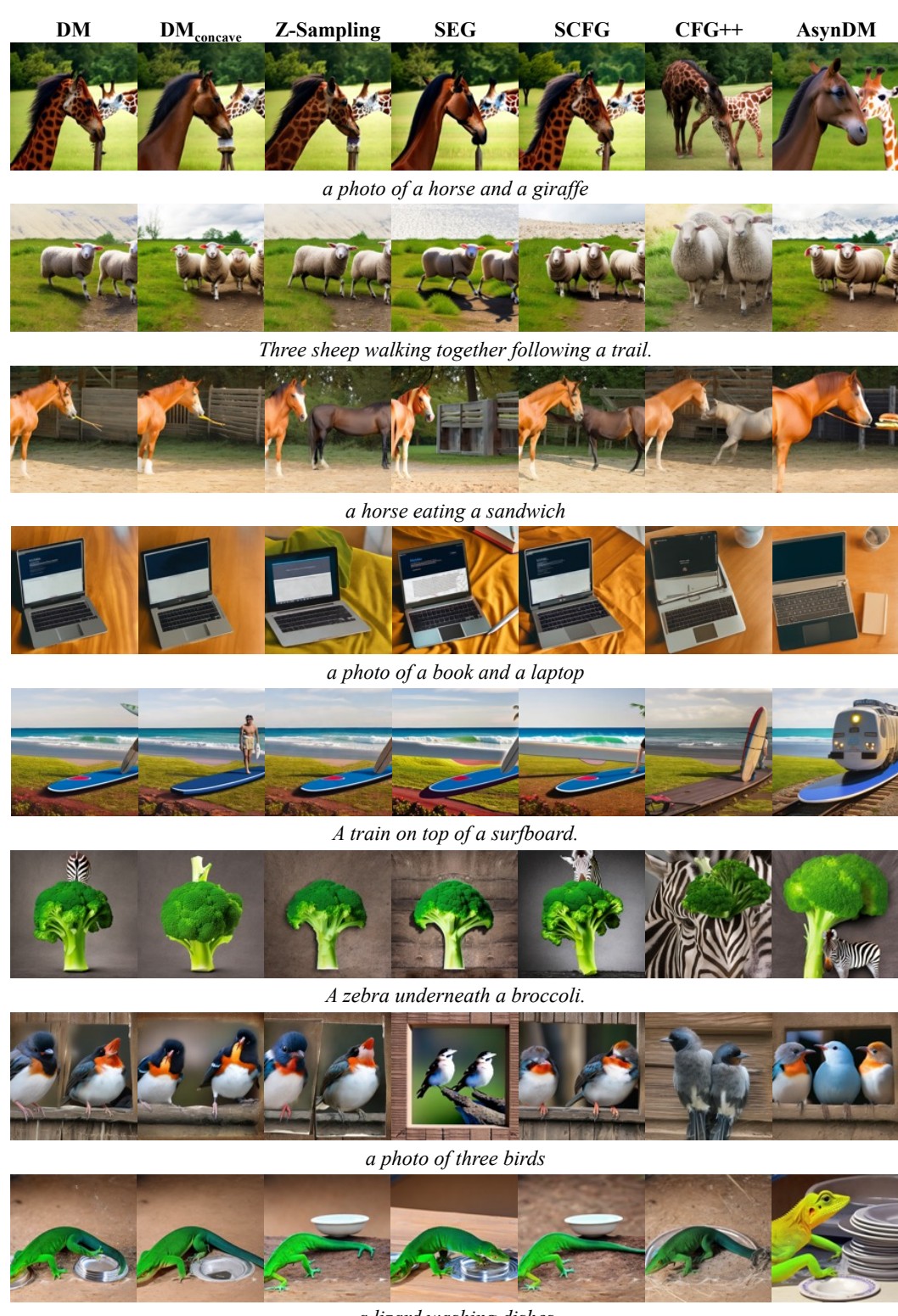

Figure 10: More samples generated by AsynDM compared with baseline methods. The images sampled by AsynDM show higher text-to-image alignment. The base model used to sample these images is SD2.1-512-base.

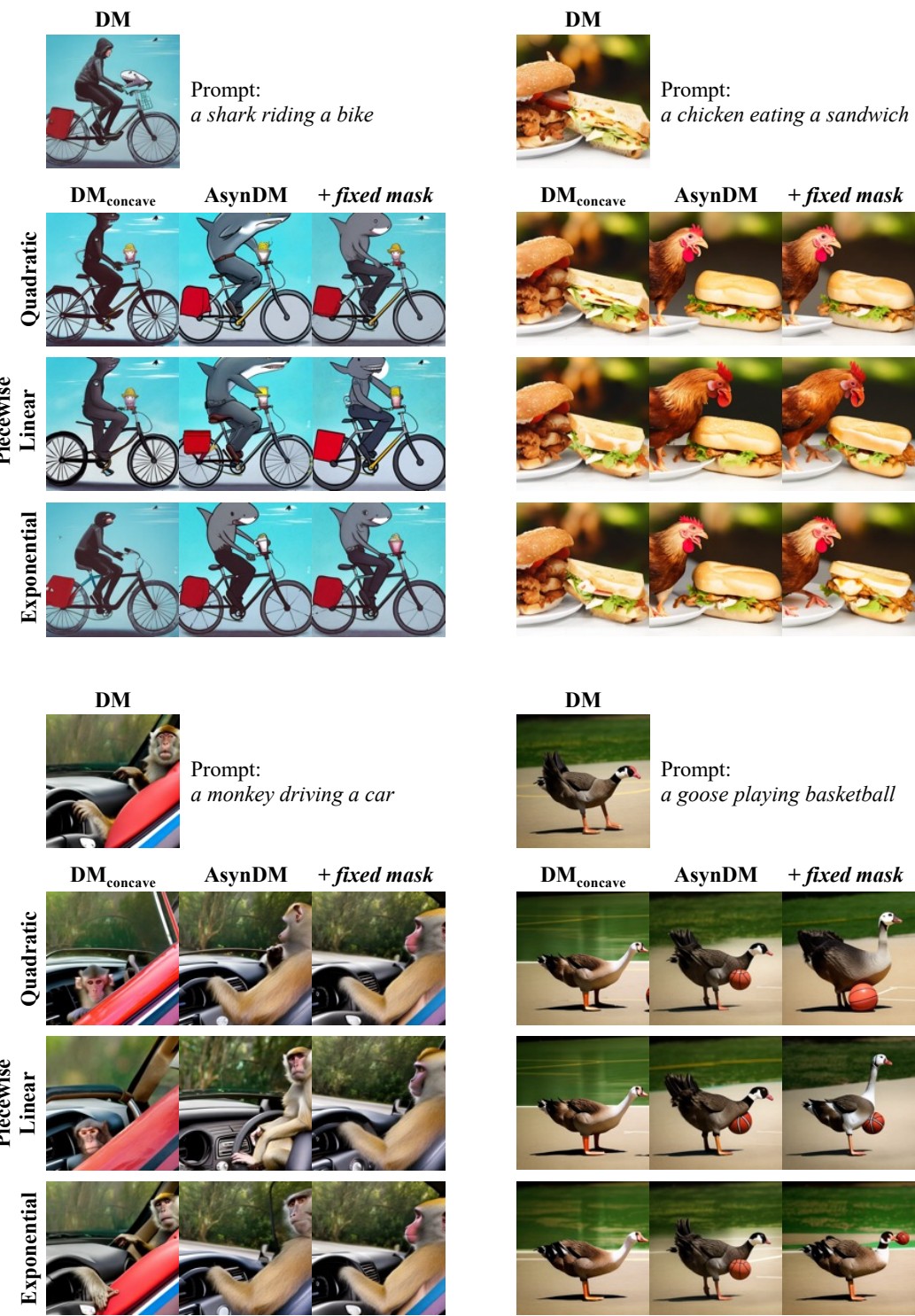

Figure 11: Samples generated by AsynDM when employing different concave schedulers and using fixed masks. The base model used to sample these images is SD2.1-512-base.

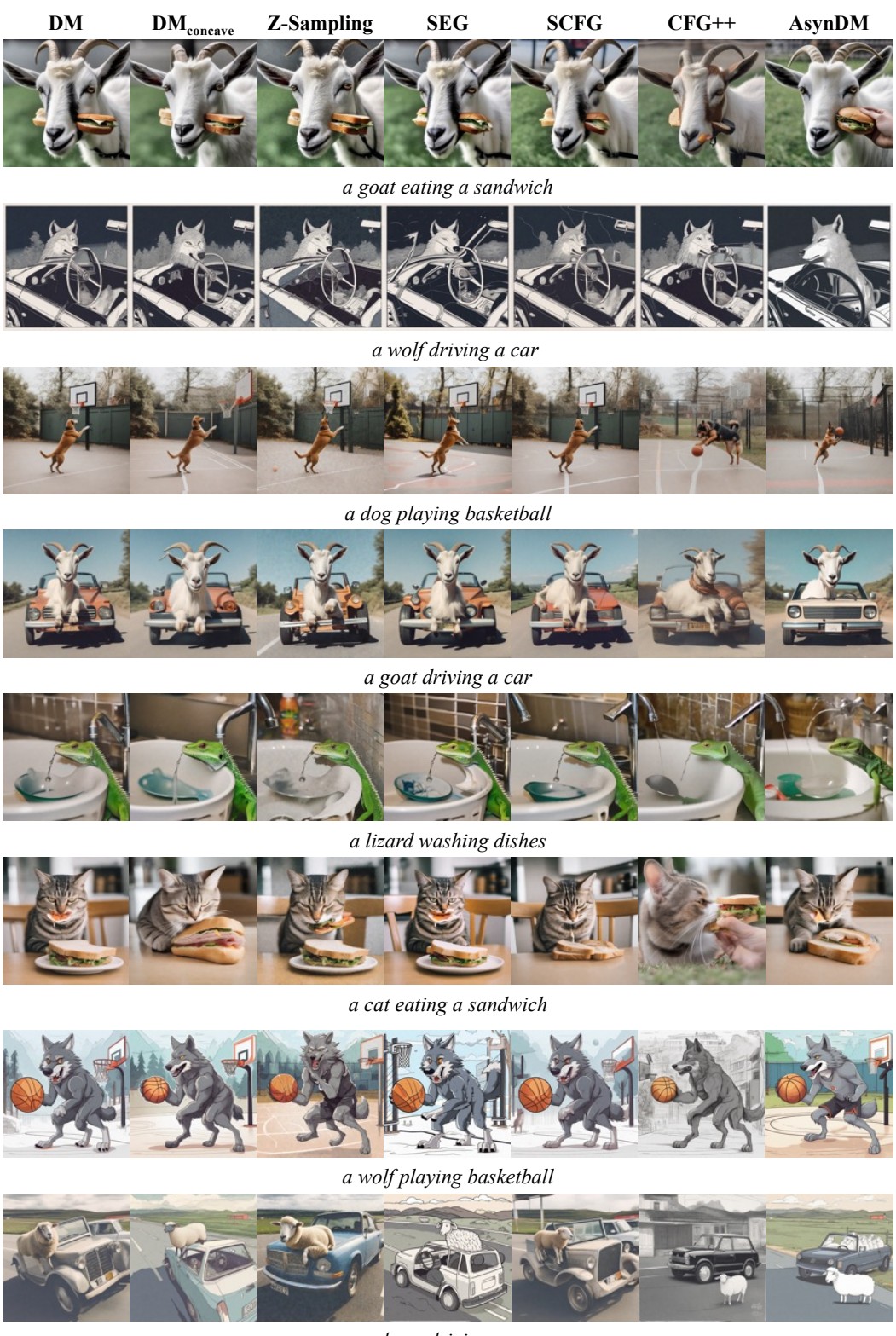

Figure 12: Samples generated by AsynDM compared with baseline methods when using SDXL-base-1.0. The images sampled by AsynDM show higher text-to-image alignment.

| DM | DM$_{concave}$ | SEG | SCFG | AsynDM |
|---|---|---|---|---|

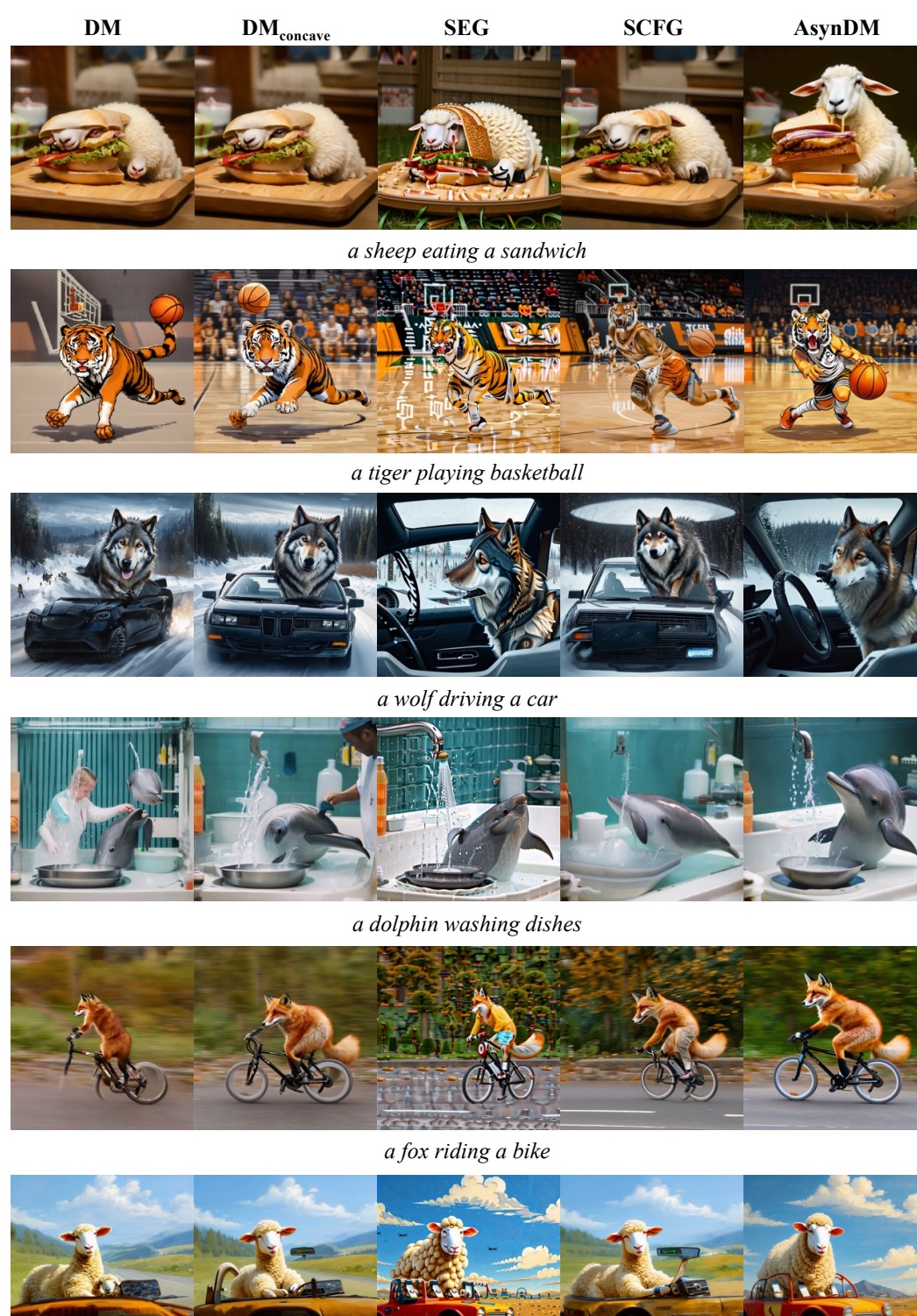

*a sheep eating a sandwich*

*a tiger playing basketball*

*a wolf driving a car*

*a dolphin washing dishes*

*a fox riding a bike*

*a sheep driving a car*

Figure 13: Samples generated by AsynDM compared with baseline methods when using SD3.5-medium. The images sampled by AsynDM show higher text-to-image alignment.

