# OpenReview forum: "Asynchronous Denoising Diffusion Models for Aligning Text-to-Image Generation"
_ICLR.cc/2026/Conference — ICLR 2026 Poster_

### Official Review · Reviewer_6tYP · 2025-10-19

**Soundness:** 4
**Presentation:** 4
**Contribution:** 3
**Rating:** 6
**Confidence:** 4

**Summary:**

This paper proposes an asynchronous diffusion framework for text-to-image generation. The core idea is to create a pixel-level timestep scheduler and let prompt-related regions be decoded more slowly. Extensive experiments demonstrate AyncDM achieves better performance among other training-free text-to-image alignment approaches.

**Strengths:**

1. The idea is novel and well-motivated. The proposed approach, using cross-attention as a mask indicator, is quite intuitive and easy to follow.
2. This paper is clearly written and well-organized.
3. The comparison results are very promising, showing clear advantages over relevant baselines.
4. The authors conduct comprehensive experiments across multiple model baselines, sampler choices, and ablation settings, which provide strong and convincing evidence for the proposed approach.

**Weaknesses:**

1. My main concern is the inconsistency between the training and inference stages of AyncDM. From my understanding, during training, noise is added synchronously into all pixels, and the diffusion model predicts $f_\theta(x_t)$ where $x_t$ is a noised latent with uniform noise levels. However, during inference, the input of the diffusion model is asynchronous or spatially varying noise levels across pixels. How can the model reliably decode latents that contain uneven noise distributions, given that it was never explicitly trained under such conditions?  Some marginal artifacts may occur in these scenarios.
2. The distracted attention mask relies heavily on cross-attention maps. However, in the early denoising steps, cross-attention maps are often noisy and unstable, which may lead to unreliable or ambiguous guidance when determining which regions should be denoised faster or slower.
3. Similarly, for more advanced T2I models such as SD3.5 or Flux that adopt the MMDiT framework instead of conventional cross-attention, deriving reliable spatial masks becomes more challenging, since text and image latents are concatenated within a self-attention module.

**Questions:**

1. How do the authors address the training–inference gap in AsynDM? Is there any theoretical or empirical evidence showing that this discrepancy can be safely ignored, or that it does not occur during sampling?
2. How does AsynDM apply or adapt the distracted cross-attention mechanism to SD3.5-Medium or other architectures based on MMDiT? The process seems non-trivial and could benefit from further clarification.

I would be happy to raise my score if these concerns are properly addressed.

---

> ### Author Response · Authors · 2025-11-17
> **Response to Reviewer 6tYP**
>
> We sincerely appreciate your careful review of our manuscript. We are delighted that you think our work well-motivated and well-organized, and we are grateful for your positive assessment of our experimental design and results.
>
> Below we provide our responses to your concerns. If any part remains unclear or if you have additional comments, please do not hesitate to let us know!
>
> ---
>
> > **Weakness 1 & Question 1**: How do the authors address the training–inference gap in AsynDM? Is there any theoretical or empirical evidence showing that this discrepancy can be safely ignored, or that it does not occur during sampling?
>
> We are pleased that you have drawn attention to this important issue. We also discussed this issue in our manuscript.
>
> As clarified in our manuscript (the first paragraph of Section 3.1), our central argument is that modern diffusion models primarily rely on the **attention module** to mediate interactions between pixels. Since the attention module does not take the timestep as an explicit input, it exhibits a certain degree of robustness when handling pixels at different timesteps.
>
> In Appendix D.2, we further investigate this phenomenon **empirically**. Specifically, we examine how large a timestep discrepancy between pixels the model can tolerate. Results show that, when the maximum discrepancy reaches approximately $0.8\times \frac{T}{2}$ or $0.9\times \frac{T}{2}$, the generated image may show a slightly blurry or noisy background (the foreground remains clear). This relatively generous tolerance allows us to perform training-free asynchronous denoising in most practical scenarios :-).
>
> ---
>
> > **Weakness 2**: In the early denoising steps, cross-attention maps are often noisy and unstable, which may lead to unreliable or ambiguous guidance when determining which regions should be denoised faster or slower.
>
> Thank you for this concern.
>
> In the early denoising steps, the cross-attention maps indeed do not provide a perfectly accurate mask. However, the **coarse structure of each object is already established at the beginning of the denoising process**, and subsequent steps mainly refine details along this initial outline. This ensures that most regions corresponding to prompt-related objects consistently denoise more slowly.
>
> Moreover, our method incorporates a dedicated mechanism-the evolving mask described in Section 4—to address the above issue. It allows the model to continuously adjust object boundaries throughout the denoising process. It is difficult to determine in advance which regions in the image are optimal for alignment. This inaccurate yet dynamic masking mechanism **offers the model opportunities to "revise its decisions"**, allowing it to adapt object boundaries according to the actual generative context. As a result, the denoising process can produce to a coherent and well-aligned final image.
>
> ---
>
> > **Weakness 3 & Question 2**: How does AsynDM apply or adapt the distracted cross-attention mechanism to SD3.5-Medium or other architectures based on MMDiT?
>
> We acknowledge that our brief mention of the implicit cross-attention mechanism in the DiT architecture in Section 4 lacked sufficient clarity, and we appreciate you for pointing this out. We have added **Appendix B.3** to provide a detailed explanation of the process. For your convenience, we also offer a brief overview here (Appendix B.3 is clearer, as it includes an illustrative figure).
>
> In the cross-attention modules, the cross-attention maps are computed using $A = \text{softmax}(\frac{QK^\top}{\sqrt{d_\text{key}}})$, where $Q$ has a shape of $(h\times w)\times d_\text{key}$, $Q$ has a shape of $|\mathbf{c}|\times d_\text{key}$, and $A$ has a shape of $(h\times w)\times |\mathbf{c}|$. However, DiT models typically do not contain cross-attention modules. Instead, they concatenate the image features and prompt embeddings, which are jointly encoded into the queries $Q$ and keys $K$, before performing the attention operation. In this setting, $Q$ and $K$ take the shape of $(h\times w+|\mathbf{c}|)\times d_\text{key}$, thus the resulting attention maps $A_\text{joint}$ has a shape of $(h\times w+|\mathbf{c}|)\times (h\times w+|\mathbf{c}|)$. By **extracting the submatrix corresponding to the interactions between the image-feature queries and the prompt-embedding keys**, we can obtain the cross-attention maps (*i.e.*, $A=A_\text{joint}[:(h\times w), (h\times w):]$, and $A$ still has a shape of $(h\times w)\times |\mathbf{c}|$). Therefore, we can also extract masks from cross-attention maps in DiT-based models.

---

> ### Author Response · Authors · 2025-11-27
> **Looking Forward to Discussion**
>
> Dear Reviewer 6tYP,
>
> We sincerely appreciate the time and effort you have dedicated to reviewing our paper.
>
> As the discussion period is drawing to a close, we would like to kindly check whether you have any remaining questions or points you would like us to clarify. We would be glad to provide any further information and are very grateful for any additional feedback you may have.
>
> Thank you very much, and we look forward to hearing from you!
>
> Best regards,
>
> Authors

---

### Official Review · Reviewer_mDhN · 2025-10-22

**Soundness:** 2
**Presentation:** 2
**Contribution:** 2
**Rating:** 2
**Confidence:** 5

**Summary:**

The work deals with text-to-image (T2I) diffusion models. The authors argue that the misalignment of the generated image and the textual prompt results from a spatially and temporally uniform (synchronous) change of the denoised image pixels. Hence they propose to denoise some regions differently, at different time steps during the reverse denoising process, the timestep of every pixel being dynamically determined. For this a timestep is assigned to each pixel. The regions of the image that are significantly associated to some tokens of the prompt are thus scheduled according to a specific (decreasing concave) function. The approach is compared to recent baseline models on four prompt datasets, reporting four alignment metrics.

**Strengths:**

* the initial motivation -- the fact that it might be worthwhile to denoise differently various regions of an image -- is intuitive and makes sense. It is clearly explained both in the introduction (section 1) and the method itself (section 3).

* the quantitative and qualitative results are reported with three different backbones, including the recent SD 3.5 (in the appendix). The proposed approach is compared to four recent baselines, published in 2024 or 2025 in top-tier venues (CVPR, NeurIPS, ICLR).

* the quantitative results are computed with a significant number of 1280 images per prompt set, with the same random seed for all models.

**Weaknesses:**

* the paper ignores an important part of the literature relating to generative semantic nursing. Following Attend-and-Excite [d] several papers investigated at optimizing alignment (cross attention) between the prompt and the noise during the backward process e.g Divide and Bind [g] or Syngen [h]. Similarly to the proposed work, these works showed that the "regions" of the image are indeed decoded at various timestep, but the conclusion was rather than the diffusion models first reconstruct the high-power, low-frequency components at early denoising stages before adding low-power, high-frequency details at later stages [i,j]. Positioning with regards to these works would thus have been relevant.

* The quantitative results are not convincing
  - the authors do not adopt previous protocol. In particular for GenEval, they use the same 553 prompt but do not use the metrics of the GenEval benchmark itself, making hard to compare to previous published results. For Drawbench, the metrics are not the same as e.g (Bai LiChen et al, 2025), making hard to compare directly with previous published results
  - the used BERTscore relies on image description obtained with an ad-hoc model, Qwen2.5-VL-7B-Instruct in this paper. The resulting scores thus evaluates both the models tested and the model used to generate the "ground truth". Similar remarks applies to the QwenScore. However, the two other metrics reflects also alignment (see below)
  - all the metrics deal with text-to-image alignment, ignoring other aspects of image generation. One can understand that this aspect is important to evaluate for this paper, but it should have been asserted that, for example, the image quality is -- at least -- maintained.
  - the quantitative results are reported without any standard deviation, while the quality of generated images (both for aesthetic and alignment) is known to be variable w.r.t the seed. Given the limited difference in performances in comparison to baselines (in particular ofor the most recent backbones in Table 4 and Table 5), one can have doubt regarding the significance.
    - by the way, it is hard to understand why the results in the main paper are based on a old model (SD 2.1) while results on more recent SDXL

* the human study is poorly described
  - there no detail on the 22 participants: are they diversified in gender and age ? Is a majority of them student of the same university as the authors? Or even colleagues? Or the author themselves? Is there some diversity in terms of native language? if so, how the prompt were presented (in English or in their native language) ?
  - it is not clear how many triplet were shown to the participants, nor how these triplets were chosen. For the automatic evaluation of Table 1 it is said that $4\times 1280$ images are considered for each of the prompt sets. Just after, on line 376, it is reported that the participant evaluate "for each group of three candidate", suggesting that they evaluate 5120 triplets. One can doubt that participant actually made as many evaluation.
  - there is no inter-annotator agreement, making hard to estimate the relevance of the study
  - for good practice on human studies, one can refer to [f]

* the qualitative results in Figure 4 are poor for the baselines, but it seems to be mainly due to the old SD 2.1 backbone used. If one uses the [online inference available for SDXL on huggingface](https://huggingface.co/stabilityai/stable-diffusion-xl-base-1.0) -- while it is itself quite old since released in July 2023 -- it is quite easy to get much better results for the base DM than those reported in Figure 4 (with the same prompts).
  - in the Appendix, on Figure 10 and 11, the qualitative results for SDXL and SD 3.5 looks often better for the baselines than the proposed model.

* minor
  - the definition of $x_i\in\mathbb{R}^{n_c\times h\times w}$ on line 194 should be introduced earlier, before equation (4) around line 184. It is indeed a crucial change for the proposed method since it reflects the *pixel-level* aspect.
  - the references for "text-to-image misalignment" on lines 057-058 are recent but inappropriate since this phenomenon has been identified well before for diffusion models, e.g in DALL-e [a] released as a preprint, Imagen (Saharia et al, 2022), DAAM [b], Structured Guidance [c] and Attend&Excite [d]. Not to mention previous works with GANs e.g SOA [e]. The reference (Liu et al, 2025) may be relevant since it is a review, although it seems to be only a preprint and not (yet?) published. However (Hu et al, 2025a) is just a recent paper dealing with a problem already known.
  - which "SD 2.1 base" (line 300) is used ? SD 2.1-512 or SD 2.1-768?

[a] A. Ramesh et al. "Hierarchical Text-Conditional Image Generation with CLIP Latents". In: arXiv 2204.06125 (2022). arXiv: 2204.0612

[b] R. Tang et al. "What the DAAM: Interpreting Stable Diffusion Using Cross Attention". ACL 2023.

[c] W. Feng et al. "Training-Free Structured Diffusion Guidance for Compositional Text-to-Image Synthesis". ICLR 2023.

[d] H. Chefer et al. "Attend-and-Excite: Attention-Based Semantic Guidance for Text-to-Image Diffusion Models". In: ACM Trans. Graph. 42.4 (July 2023)

[e] Hinz et al "Semantic Object Accuracy for Generative Text-to-Image Synthesis", TPAMI 2022 (and arXiv:1910.13321 in 2019)

[f] M. Otani et al. "Toward Verifiable and Reproducible Human Evaluation for Text-to-Image Generation". CVPR, 2023

[g] Li et al "Divide & Bind Your Attention for Improved Generative Semantic Nursing", BMVC 2023

[h] Rassin et al "Linguistic Binding in Diffusion Models: Enhancing Attribute Correspondence through Attention Map Alignment" ,NeurIPS 2023

[i] S. Rissanen, M. Heinonen, and A. Solin. “Generative modelling with inverse heat dissipation”. ICLR 2023

[j] Y.-H. Park et al. “Understanding the Latent Space of Diffusion Models through the Lens of Riemannian Geometry”. NeurIPS 2023

**Questions:**

- which "SD 2.1 base" (line 300) is used ? SD 2.1-512 or SD 2.1-768?
- why the results on SDXL and SD 3.5  have not been reported in the main paper (and those with SD 2.1 in appendix) ?
- Could we have more details about the study conducted with humans, in particular on the cohort of 22 participants (see above)?

**Details Of Ethics Concerns:**

Ethic concern is not addressed. Since the work can be applied to image synthesis in particular, there are many potential societal consequences of the work. One may, as (Esser et al, 2024) point to [k] for a discussion on these.

[k] Po, R.et al "State of the Art on Diffusion Models for Visual Computing". Computer Graphics Forum, 43: e15063. 2024

---

> ### Author Response · Authors · 2025-11-17
> **Response to Reviewer mDhN (Part 1/3)**
>
> We sincerely thank you for the time and effort you dedicated to reviewing our paper and for providing such detailed comments. We are pleased that you think our work well-motivated and that you recognize our main experimental settings.
>
> Below we address your concerns point by point. If something remains unclear or you have additional insights, please do not hesitate to let us know!
>
> ---
>
> > **Weakness 1**: The paper ignores an important part of the literature relating to generative semantic nursing.
>
> Thank you for drawing our attention to the line of work on generative semantic nursing. Previously, our background Section 2.2 focused primarily on model fine-tuning and tuning-free approaches. We have now incorporated this body of research as a separate category.
>
> We also appreciate your recommendation of works [i, j], and we have carefully integrated their insights into the revised manuscript.
>
> ---
>
> > **Weakness 2-1**: The authors do not adopt previous protocol for GenEval and Drawbench.
>
> Thank you for raising this concern. We understand that your main concern is that if our evaluation metrics differ from those used in prior work, it may be difficult to directly compare our results with previously published ones. We would like to clarify the following points:
> * Different works typically rely on different metrics, and it is often impractical to reproduce all of them for comparison.
> * Even when using the same metric, **direct comparison with previously published results is still challenging**. Variations in experimental settings (*e.g.*, denoising steps, base models) can affect the scores, yet these settings are rarely identical across studies.
> * Unlike deterministic metrics such as accuracy, most **image metrics rely on model prediction**. As model versions evolve, these metrics naturally improve over time, so adopting more advanced metrics should be acceptable.
>
> To ensure the reliability of our evaluation, we selected four metrics and evaluated all methods **under identical experimental settings**. Moreover, our tests were conducted across four prompt sets. We believe that the results obtained under this evaluation settings are valid and reliable.
>
> ---
>
> > **Weakness 2-2**: The used BERTscore relies on image description obtained with an ad-hoc model, Qwen2.5-VL-7B-Instruct in this paper.
>
> On one hand, the evaluation pipeline based on BERTScore was first introduced by [1] (the original version employs LLaVA to generate image captions, while we replaced it with a more advanced model, Qwen2.5-VL-7B-Instruct). This pipeline has been widely adopted in subsequent studies (*e.g.*, [2]). On the other hand, directly using multimodal large language models (MLLMs) to score images is also a common practice in recent works (*e.g.*, [3]).
>
> We understand the concern that current MLLMs may exhibit hallucinations. However, researchers generally agree that current models exhibit **stronger image understanding capabilities than image generation capabilities** (*e.g.*, [4]). Therefore, leveraging MLLMs to evaluate image generation quality is considered acceptable and reasonable.
>
> [1] Training Diffusion Models with Reinforcement Learning, ICLR 2024.
>
> [2] Decoding Correlation-Induced Misalignment in the Stable Diffusion Workflow for Text-to-Image Generation, ICCV 2025.
>
> [3] Noise Diffusion for Enhancing Semantic Faithfulness in Text-to-Image Synthesis, CVPR 2025.
>
> [4] HermesFlow:  Seamlessly Closing the Gap in Multimodal Understanding and Generation, NeurIPS 2025.
>
> ---
>
> > **Weakness 2-3**: All the metrics deal with text-to-image alignment, ignoring other aspects of image generation, for example, the image quality.
>
> Thank you for raising this point. Evaluating image quality is indeed crucial, as it demonstrates that our method does not improve alignment at the expense of quality. To address this, we additionally report FID-30K results (FID-30K refers to the Frechet Inception Distance calculated using 30,000 images from the MSCOCO 2024 validation set as the reference dataset).
>
> Since FID evaluation typically requires tens of thousands of samples, the 1,280 images used for our other metrics are insufficient. Therefore, we merge all four prompt sets and generate 16,000 images with each model. The resulting scores are shown below. These results indicate that our method can largely preserve the image quality of the pretrained diffusion model. We also added the results to Section 5.3.
>
> | Model | FID-30K$\downarrow$ |
> |---|---|
> | DM | 48.63 |
> | $\text{DM}_\text{concave}$ | 49.29 |
> | AsynDM | 49.38 |

---

> ### Author Response · Authors · 2025-11-17
> **Response to Reviewer mDhN (Part 2/3)**
>
> > **Weakness 2-4**: The quantitative results are reported without any standard deviation. One can have doubt regarding the significance.
>
> Our work, as well as similar baseline studies, did not report the standard deviation. We would like to clarify this choice as follows:
> * **The alignment of images generated with different random seeds varies considerably**. Whether in image or text generation, such variations in outputs are common in large-scale pretrained models. (Notably, this variability is also the foundation for assigning different rewards to different outputs in large model reinforcement learning.)
> * **A standard deviation larger than the mean improvement does not necessarily imply that the improvement is insignificant**. While certain seeds may occasionally yield higher scores, repeatedly sampling with multiple seeds would incur considerable computational overhead. The main value of our approach lies in achieving higher alignment regardless of the specific random seed used.
> * **The improvement can be reasonably demonstrated by comparing mean scores**, since all methods are evaluated under the same random seeds.
> * **Regarding the base model**. SDXL and SD3.5 exhibit noticeably better alignment than SD2.1. However, this does not diminish the value of our work. Our goal is to introduce a lightweight and compatible method that can be applied across different base models. Despite the rapid progress of pretrained models, this line of research continues to hold importance.
>
> Based on the above facts, standard deviation is not an effective evaluation protocol for capturing text-to-image alignment. Nonetheless, if you would like to examine this statistics, we are glad to attach them in the appendix further.
>
> ---
>
> > **Weakness 3 & Question 3**: The human study is poorly described.
>
> Thank you for your detailed comments and for recommending a valuable reference. We have added more details about the human study in **Appendix B.2**, which are also presented below. Due to limited resources, we were only able to recruit 22 participants, which inevitably makes it difficult to cover all demographic groups. However, such participant numbers are common in similar studies (*e.g.*, [1] includes 8 participants and [2] includes 5). We hope for your kind understanding.
>
> * Appendix B.2: The human study was conducted with 22 participants from five universities, including 10 females and 12 males, whose academic backgrounds ranged from undergraduates to Ph.D. students. Each participant was asked to complete a form. At the beginning of the form, we provided the following instruction: "*Image–prompt alignment refers to how well an image matches the given textual description. For each group of three images, please select the one you believe best matches the given textual description.*" The form contained 64 groups of images in total, corresponding to four prompt sets, each comprising 16 groups. These 64 groups were randomly selected from the images sampled during the evaluation of results reported in Table 1. Each group consisted of three images generated by DM, $\text{DM}_\text{concave}$ and AsynDM under the same random seed, accompanied by the corresponding text prompt used for generation. All participants received the same set of images, but the presentation order was randomized, ensuring that participants were unaware of which method each image originated from. The entire form was presented in English.
>
> [1] DPOK: Reinforcement Learning for Fine-tuning Text-to-Image Diffusion Models, NeurIPS 2023.
>
> [2] Rethinking the Spatial Inconsistency in Classifier-Free Diffusion Guidance, CVPR 2024.

---

> ### Author Response · Authors · 2025-11-17
> **Response to Reviewer mDhN (Part 3/3)**
>
> > **Weakness 4**: If one uses the SDXL, it is quite easy to get much better results for the base DM than those reported in Figure 4 (with the same prompts).
> >
> > **Question 2**: Why the results on SDXL and SD 3.5 have not been reported in the main paper (and those with SD 2.1 in appendix)?
>
> Although SDXL and SD3.5 are more advanced than SD2.1, we chose SD2.1 as the primary model in our experiments for two main reasons.
> * **Reason 1**: As you mentioned, SDXL can easily outperform SD2.1 with the prompts used in our paper. However, it is undeniable that SDXL also suffers from misalignment issues, particularly when dealing with long and complex prompts. Currently, there is no widely recognized **benchmark of complex prompts** suitable for evaluating advanced models, and existing **metrics** may not accurately capture alignment performance when using complex prompts. Therefore, using SD2.1 allows us to study this problem in a more evident manner.
> * **Reason 2**: Our computational resources are limited, preventing us from conducting experiments on SDXL and SD3.5 with the same level of thoroughness as on SD2.1.
>
> While more advanced models may directly produce images with higher alignment than those generated by our method on SD2.1, we believe that our work remains meaningful. This is because our work provides a **compatible** and **low-overhead** approach that can be readily applied to different diffusion models. As shown in the paper, for more advanced models, AsynDM can further enhance their alignment performance as well.
>
> ---
>
> > **Weakness 5-1**: The definition of $x_i$ on line 194 should be introduced earlier.
>
> Thank you for this valuable suggestion, and we have revised the manuscript accordingly :-).
>
> ---
>
> > **Weakness 5-2**: The references for "text-to-image misalignment" on lines 057-058 are recent but inappropriate.
>
> Thank you for recommending these more appropriate references. Misalignment has been a long-standing research topic. Yet, in earlier works, it was often referred to by different terms (*e.g.*, text-image inconsistency), which is why we originally cited more recent papers. We agree that the papers you suggested are more suitable and thoughtful, and we have now included them in the corresponding sections.
>
> ---
>
> > **Weakness 5-3 & Question 1**: which "SD 2.1 base" (line 300) is used? SD 2.1-512 or SD 2.1-768?
>
> We used **SD 2.1-512** in our experiments. In fact, SD 2.1-base refers to the 512-resolution version, while SD 2.1-v corresponds to the 768-resolution version (see [Model Card](https://huggingface.co/stabilityai/stable-diffusion-2-1)). Nonetheless, we appreciate your attention to this detail and have revised the paper accordingly to make it clear.
>
> ---
>
> > **Ethics Concerns**: Ethic concern is not addressed. Since the work can be applied to image synthesis in particular, there are many potential societal consequences of the work.
>
> We thank you for the reminder and for suggesting a valuable reference. To address the relevant concerns, we have included an **ethics statement** before the reproducibility statement.

---

> ### Comment · Reviewer_mDhN · 2025-11-20
>
> # Weakness 2-4: The quantitative results are reported without any standard deviation
> * the fact that "The alignment of images generated with different random seeds varies considerably" is precisely the reason why standard deviation *should* be reported
> * "A standard deviation larger than the mean improvement does not necessarily imply that the improvement is insignificant. "  I agree that a **small standard deviation is not enough** to assert properly the significance of results. You may also have significant results with large variance.  A proper mean would be to use statistical tests but it leads to other issues (choosing the right one...). In any case, ""A standard deviation larger than the mean improvement" **is a strong hint that the results may be not significant**. Thus, yes, I would be very interested in having the variation of results for the proposed approach and baselines.
> * " SDXL and SD3.5 exhibit noticeably better alignment than SD2.1 (...) [but] does not diminish the value of our work." Actually, it is often easier to have good results with less performing approaches. Thus reporting the results on state-of-the -art models such as SD3.5 has a higher value. To clarify: at the opposite, how would a reviewer judge a work that exhibits better results on conditional GAN (2014). While much "lightweight", would it have any value in 2026 ? obviously SD2.1 is much closer from the state of the art than cGAN but the idea is that comparing on the best available models is often more convincing
>
> Also note that if the significance is interesting for Table 1, it is even more important in Table 2 to show that AsynDM has an interest over DM.
>
> Last, let note that in the experiment above, DM has a FID of 48.63 while the proposed approach has one of 49.38. You conclude that your approach "can largely preserve the image quality" thus that the difference is not significant. Yet, the difference is larger that any improvement reported in Table 1 in terms of BERTscore of CLIPscore...
>
> # Weakness 3 & Question 3: The human study is poorly described.
> I acknowledge the details provided. It nevertheless misses the most important one, namely the inter-annotator agreement. To which extend the annotator agree on the annotations ? Since "All participants received the same set of images" it should be easy to compute it e.g  Fleiss’ kappa.
>
> Regarding the number of participants, the fact that previous papers reported human studies with not enough persons, that mainly show that their reviewers lacked expertise on the subject of human studies, is not a reason to accept new badly design ones. Following (Otani et al, 2023) the idea is to motivate the community to improve their practice. Even before that paper (Chefer et al. 2022) conducted their user study with 65 persons. However, beyond the number of humans, their inter-annotator agreement is the most important

---

> > ### Author Response · Authors · 2025-11-21
> >
> > Thank you for your active feedback. Here, we provide further response and clarification to address your concern.
> >
> > ---
> >
> > > **About Standard Deviation**: I would be very interested in having the variation of results for the proposed approach and baselines.
> >
> > We have reported the standard deviations of the results in Table 1, Table 2, Table 4, and Table 5 in **Appendix D.4**. As mentioned above, these standard deviations are larger than the average improvements.
> >
> > Achieving improvements that consistently exceed the standard deviation is challenging. Due to the training-free nature, both our method and the baselines **operate within the model's existing capacity, without altering its intrinsic performance ceiling**. Our method strives to more fully exploit the capabilities of the pretrained model to deliver better alignment.
> >
> > Besides, we note that neither the baselines nor the work you referenced use standard deviation as a criterion for evaluating their target objectives, and most of them do not report it. Other reviewers also pointed out that alignment (and other image-quality properties) cannot be deterministically assessed by standard deviation. Taken together, these observations give us sufficient reason to believe that the community generally tends to report the improvement metrics rather than relying on standard deviation.
> >
> > ---
> >
> > > **About Base Model**: Thus reporting the results on state-of-the-art models such as SD3.5 has a higher value. To clarify: at the opposite, how would a reviewer judge a work that exhibits better results on conditional GAN (2014). While much "lightweight", would it have any value in 2026 ?
> >
> > We believe this analogy does not accurately capture our scenario. Contemporary research pays little attention to cGANs because GANs are no longer the state-of-the-art paradigm for image generation. Diffusion models, however, remain the dominant and actively studied paradigm. Our method is compatible across different variants of diffusion models and therefore has clear potential for practical applications.
> >
> > Moreover, we have conducted experiments on both SDXL and SD3.5. The reason why these results are reported in appendix has already been explained in our **response to Weakness 4 and Question 2**.
> >
> > ---
> >
> > > **About FID**: Yet, the difference in FID is larger that any improvement reported in Table 1 in terms of BERTscore of CLIPscore...
> >
> > FID measures image quality / fidelity and is defined on a range from 0 to positive infinity. In contrast, BERTScore and CLIPScore evaluate prompt–image alignment, with value ranges from 0 to 1 and from –1 to 1 respectively. As these metrics quantify fundamentally different properties and operate on incompatible scales, comparing them directly in terms of numerical values is inappropriate and unreasonable.
> >
> > ---
> >
> > > **About Human Study**: It nevertheless misses the most important one, namely the inter-annotator agreement.
> > > Regarding the number of participants, ...
> >
> > Your extensive experience with human studies has provided us with valuable feedback. Following your suggestions, we **increased the number of human participants to 52** and updated the corresponding evaluation results in the paper. Additionally, as you recommended, we computed Fleiss' kappa, which yielded a value of **0.4252**. It is well within the normal range for inter-annotator agreement.

---

> ### Author Response · Authors · 2025-11-27
> **Looking Forward to Discussion**
>
> Dear Reviewer mDhN,
>
> Thank you sincerely for the thoughtful attention you have given to our submission.
>
> With the discussion phase nearing its end, we wanted to reach out and see if there are any remaining concerns or points you would like us to address. We would be more than happy to provide further clarification or additional details if needed, and we truly appreciate any further comments you may wish to share.
>
> Many thanks again!
>
> Best regards,
>
> Authors

---

### Official Review · Reviewer_QfCK · 2025-10-29

**Soundness:** 3
**Presentation:** 3
**Contribution:** 3
**Rating:** 6
**Confidence:** 4

**Summary:**

This paper proposes a novel method that utilizes masks (getting from network attention or using fixed masks) to decompose the diffusion time of every pixels in an image, resulting clearer inter-pixel context and significant improvement.

**Strengths:**

**S1: Very good innovation.** I believe asynchronous diffusion can bring a lot of inspiration to subsequent work.

**S2: Significant improvement.** Figure 4 (I was looking forward to discovering more in the supplementary materials, but I couldn't find them) and human survey show extremely good improvements.

**Weaknesses:**

**W1: Claim issue**. In intro., the authors claim that the text-to-image misalignment is caused primarily by synchronous denoising. However, in many other single-step generative models, including GANs and VAEs, the misalignment also is a key problem. Thus, I think this statement lacks strong support. In other words, I believe that the proposed asynchronous denoising method can alleviate this misalignment problem to some extent, but this misalignment may not necessarily be caused by this synchronous denoising. Therefore, I believe that this statement needs to be revised and the author needs to provide a broader discussion on other methods to address this issue (including discussing existing methods [1] [2] [3] in other generative models to address this issue and other potential solutions).

[1] Liao W, Hu K, Yang M Y, et al. Text to image generation with semantic-spatial aware gan[C]//Proceedings of the IEEE/CVF conference on computer vision and pattern recognition. 2022: 18187-18196.

[2] Zhang C, Peng Y. Stacking VAE and GAN for context-aware text-to-image generation[C]//2018 IEEE Fourth International Conference on Multimedia Big Data (BigMM). IEEE, 2018: 1-5.

[3] Wang H, Lin G, Hoi S C H, et al. Cycle-consistent inverse GAN for text-to-image synthesis[C]//Proceedings of the 29th ACM international conference on multimedia. 2021: 630-638.

**Questions:**

Q1:
This proposed method uses different time steps for different pixels. However, current research on diffusion focuses on reducing the time step of diffusion. Therefore, when this method is applied to models with small-time diffusion, e.g., T=4 in DDGAN [4], can the effect still be significantly improved?

Q2:
This proposed method can be further combined with patchDiff [5] to improve the generation effect?

Q3:
From the Fig.4, we can find there are clear improvement between the proposed methods and other methods. However, the quantitative performance improvement in Table1 are not obvious. I believe this is a question worth explaining.

[4] Xiao Z, Kreis K, Vahdat A. Tackling the Generative Learning Trilemma with Denoising Diffusion GANs[C]//International Conference on Learning Representations.

[5] Wang Z, Jiang Y, Zheng H, et al. Patch diffusion: Faster and more data-efficient training of diffusion models[J]. Advances in neural information processing systems, 2023, 36: 72137-72154.

---

> ### Author Response · Authors · 2025-11-17
> **Response to Reviewer QfCK**
>
> We sincerely appreciate your time in reviewing our paper. We are also delighted that you think our work is innovative and that you agree with the effectiveness of our experimental results.
>
> Below are the responses to your concerns. If something remains unclear or you have further questions, please do not hesitate to let us know!
>
> ---
>
> > **Strength 2**: I was looking forward to discovering more in the supplementary materials, but I couldn't find them.
>
> We sincerely appreciate that you found Figure 4 to show clear improvements. Additional image examples can be found in Appendix E, where Figure 10 is based on SD 2.1, Figure 12 on SDXL, and Figure 13 on SD 3.5. Thank you very much for reviewing them :-).
>
> ---
>
> > **Weakness 1**: I believe that this statement needs to be revised and the author needs to provide a broader discussion on other methods to address this issue.
>
> Thank you very much for this suggestion. We agree that we can only claim that **synchronous denoising tends to produce misaligned images in diffusion models**, rather than claiming that misalignment is primarily caused by synchronous denoising. We have revised both the abstract and introduction accordingly.
>
> We also appreciate the papers you recommended. In Section 2.2 of the background, we have expanded the discussion to include GAN- and VAE-based approaches, and we have incorporated more techniques designed to mitigate misalignment.
>
> ---
>
> > **Question 1**: When this method is applied to models with small-time diffusion, e.g., T=4 in DDGAN, can the effect still be significantly improved?
>
> Thank you for raising this concern, as it is indeed an important point. Therefore, we added **Appendix D.3** to highlight that works such as DDGAN focus on generating high-quality images with only a few denoising steps. We also conducted an **ablation study** to evaluate our method under different numbers of denoising steps. The full results can be found in Appendix D.3, and we present the BERTScore evaluation here. As shown, our method continues to improve alignment even when using a small number of denoising steps.
>
> | Model | T=5 | T=10 | T=20 | T=30 | T=40 | T=50 | T=60 |
> |---|---|---|---|---|---|---|---|
> | DM | 0.5924 | 0.6221 | 0.6311 | 0.6330 | 0.6371 | 0.6353 | 0.6346 |
> | AsynDM | **0.5987** | **0.6280** | **0.6364** | **0.6372** | **0.6402** | **0.6414** | **0.6412** |
>
> ---
>
> > **Question 2**: This proposed method can be further combined with patchDiff to improve the generation effect?
>
> Thank you for recommending this work. PatchDiff is a representative label-conditioned generative method, and we have discussed it in the background section in the new version. After reading the paper, we believe our method is in principle **compatible** with it. However, despite its superiority on label-conditioned generation, we note that PatchDiff does **not support text-conditioned generation**, making it difficult to use as a base model for evaluating the effect.
>
> ---
>
> > **Question 3**: The quantitative performance improvement in Table 1 are not obvious. I believe this is a question worth explaining.
>
> Thank you for this question.
>
> The relatively small quantitative improvements are not due to insignificant gains, but rather because **small changes in these metrics can correspond to large differences in alignment**. Similar observations have been reported in prior studies focusing on enhancing text-to-image alignment. For example, [1] explicitly mentions this property of BERTScore in their Section 6.2, while Table 7 in [2] and Table 2 in [3] also report relatively minor increases in CLIPScore.
>
> With this in mind, we adopted four evaluation metrics and tested on four prompt sets to make our evaluation results more convincing and comprehensive. We hope this clarification helps convey the fairness and validity of our experimental results.
>
> [1] Training Diffusion Models with Reinforcement Learning, ICLR 2024.
>
> [2] Zigzag Diffusion Sampling: Diffusion Models Can Self-Improve via Self-Reflection, ICLR 2025.
>
> [3] Rethinking the Spatial Inconsistency in Classifier-Free Diffusion Guidance, CVPR 2024.

---

> ### Comment · Reviewer_QfCK · 2025-11-21
>
> Thanks for authors' reply.
>
> The authors' interpretation of the metrics is reasonable and consistent with the image generation community's criticism of qualitative metrics (e.g., [1], which also criticizes the clip score for not accurately reflecting the alignment effect).
>
> The experiments in this work comprehensively have covered all general metrics of image generation task; the reported standard deviation and the imperfections of human evaluation are negligible.
>
> Thus, **I believe this work fully meets ICLR's 8-point standard: accept, good paper (poster).**
>
> [1] Semantic similarity distance: Towards better text-image consistency metric in text-to-image generation, Pattern Recognition 144, 109883.

---

> > ### Author Response · Authors · 2025-11-21
> > **Thanks for Reviewing Our Paper**
> >
> > We sincerely appreciate your active engagement and thoughtful feedback in the rebuttal process. We are also truly grateful for your recognition of our work.
> >
> > Thank you once again for the time and effort you devoted to reviewing our paper!

---

### Author Response · Authors · 2025-11-17
**General Response to Reviewers**

We sincerely appreciate the time and effort each reviewer devoted to evaluating our submission. We are especially grateful that the reviewers found our paper well-motivated (Reviewers QfCK, mDhN, and 6tYP) and well-organized (Reviewer 6tYP), and that our experimental design (Reviewers mDhN and 6tYP) and experimental results (Reviewers QfCK and 6tYP) received recognition.

In response to the valuable feedback provided by the reviewers, we have made the following major **revisions** to the manuscript (highlighted in blue):
* Presentations
    * We **revised several unclear or inappropriate expressions** in the Abstract, the second paragraph of Introduction, and the second paragraph of Section 3.1.
    * We added more **related work** to Section 2.2 (Background).
* Experiments
    * We added the **FID evaluation results** in Section 5.3.
    * We added a new **ablation study** on denoising steps in Appendix D.3.
* Appendices
    * We added an **ethics statement** before the reproducibility statement.
    * We added a new **Appendix B.2** describing the details of our human study.
    * We added a new **Appendix B.3** explaining how to extract cross-attention masks from DiT-based models.

For all weaknesses and questions raised by the reviewers, we have provided point-by-point responses following their comments. We thank the reviewers again for their thoughtful feedback and warmly welcome any further questions.

---

### Meta-Review · Area_Chair_DzWn · 2026-01-06

**Summary:**

While the reviewers acknowledge the interest of the problem tackled by this work and of the proposed solution, they express concerns related to some claims made by the authors, the applicability of the proposed method within other models, some aspects of the quantitative evaluation, the discussion of related work, some aspects of the presentation, and some specific technical aspects.

**Reviewer Concerns:**

The authors provided thorough answers, which convincingly address most of the reviewers' concerns. Some points nonetheless remain debatable, such as the questions related to statistical significance, and it is unclear if the authors' answers would have fully convinced Reviewer mDhN.

**Reviewer Scores:**

In the discussion, Reviewer QfCK stated that the work deserves a score of 8. It seems likely that Reviewer 6tYP would have at least maintained their positive score of 6, if not increased it, as the authors' answers to their questions were convincing. It is, however, unclear whether Reviewer mDhN would have raised their score at all, let alone to the point of recommending acceptance. This being said, the AC argues that the authors convincingly discussed the concerns raised by the reviewer, and that there remains no truly major concerns.

---

### Decision · Program_Chairs · 2026-01-26

Accept (Poster)